# Minimalistic Unsupervised Representation Learning with the Sparse Manifold Transform

**Yubei Chen**[1,2]**, Zeyu Yun**[4,5]**, Yi Ma**[4]**, Bruno Olshausen**[4,5,6]**, Yann LeCun**[1,2,3]

[1] Meta AI
[2] Center for Data Science, [3] Courant Institute, New York University
[4] EECS Dept., [5] Redwood Center, [6] Helen Wills Neuroscience Inst., UC Berkeley

## Abstract

We describe a minimalistic and interpretable method for unsupervised representation learning that does not require data augmentation, hyperparameter tuning, or other engineering designs, but nonetheless achieves performance close to the state-of-the-art (SOTA) SSL methods. Our approach leverages the sparse manifold transform [21], which unifies sparse coding, manifold learning, and slow feature analysis. With a one-layer deterministic (one training epoch) sparse manifold transform, it is possible to achieve 99.3% KNN top-1 accuracy on MNIST, 81.1% KNN top-1 accuracy on CIFAR-10, and 53.2% on CIFAR-100. With simple grayscale augmentation, the model achieves 83.2% KNN top-1 accuracy on CIFAR-10 and 57% on CIFAR-100. These results significantly close the gap between simplistic "white-box" methods and SOTA methods. We also provide visualization to illustrate how an unsupervised representation transform is formed. The proposed method is closely connected to latent-embedding self-supervised methods and can be treated as the simplest form of VICReg. Though a small performance gap remains between our simple constructive model and SOTA methods, the evidence points to this as a promising direction for achieving a principled and white-box approach to unsupervised representation learning, which has potential to significantly improve learning efficiency.

## 1 Introduction

Unsupervised representation learning (aka self-supervised representation learning) aims to build models that automatically find patterns in data and reveal these patterns explicitly with a representation. There has been tremendous progress over the past few years in the unsupervised representation learning community, and this trend promises unparalleled scalability for future data-driven machine learning. However, questions remain about what exactly a representation *is* and how it is formed in an unsupervised fashion. Furthermore, it is unclear whether there exists a set of common principles underlying all these unsupervised representations.

Many investigators have appreciated the importance of improving our understanding of unsupervised representation learning and taken pioneering steps to simplify SOTA methods [18; 19; 114; 22], to establish connections to classical methods [69; 4], to unify different approaches [4; 39; 97; 62; 55], to visualize the representation [9; 116; 15], and to analyze the methods from a theoretical perspective [3; 45; 107; 4]. The hope is that such understanding will lead to a theory that enables us to build simple, fully explainable "white-box" models [14; 13; 71] from data based on first principles. Such a computational theory could guide us in achieving two intertwined fundamental goals: modeling natural signal statistics, and modeling biological sensory systems [83; 31; 32; 65].

Here, we take a small step toward this goal by building a minimalistic white-box unsupervised learning model without deep networks, projection heads, augmentation, or other similar engineering designs. By leveraging the classical unsupervised learning principles of sparsity [81; 82] and low-rank spectral embedding [89; 105], we build a two-layer model that achieves non-trivial benchmark results on several standard datasets. In particular, we show that a two-layer model based on the sparse manifold transform [21], which shares the same objective as latent-embedding SSL methods [5],

achieves 99.3% KNN top-1 accuracy on MNIST, 81.1% KNN top-1 accuracy on CIFAR-10, and 53.2% on CIFAR-100 without data augmentation. With simple grayscale augmentation, it achieves 83.2% KNN top-1 accuracy on CIFAR-10 and 57% KNN top-1 accuracy on CIFAR-100. These results close the gap between a white-box model and the SOTA SSL models [18; 20; 5; 117]. Though the gap remains, narrowing it further potentially leads to a deeper understanding of unsupervised representation learning, and this is a promising path towards a useful theory.

We begin the technical discussion by addressing three fundamental questions. In Section 2, we then revisit the formulation of SMT from a more general perspective and discuss how it can solve various unsupervised representation learning problems. In Section 3, we present benchmark results on MNIST, CIFAR-10, and CIFAR-100, together with visualization and ablations. Additional related literature is addressed in Section 4, and we offer further discussion in Section 5. This paper makes four main contributions. 1) The original SMT paper explains SMT from only a manifold learning perspective. We provide a novel and crucial interpretation of SMT from a probabilistic co-occurrence point of view. 2) The original SMT paper potentially creates the misleading impression that time is necessary to establish this transformation. However, this is not true. In this paper, we explain in detail how different kinds of localities (or similarities) can be leveraged to establish the transform. 3) We provide benchmark results that support the theoretical proposal in SMT. 4) We provide a connection between SMT and VICReg (and other SSL methods). Because SMT is built purely from neural and statistical principles, this leads to a better understanding of self-supervised learning models.

**Three fundamental questions:**

**What is an unsupervised (self-supervised) representation?** Any non-identity transformation of the original signal can be called a representation. One general goal in unsupervised representation learning is to find a function that transforms raw data into a new space such that "similar" things are placed closer to each other and the new space isn't simply a collapsed trivial space[1]. That is, the important geometric or stochastic structure of the data must be preserved. If this goal is achieved, then naturally "dissimilar" things would be placed far apart in the representation space.

**Where does "similarity" come from?** "Similarity" comes from three classical ideas, which have been proposed multiple times in different contexts: 1) temporal co-occurrence [112; 119], 2) spatial co-occurrence[90; 28; 34; 103; 84], and 3) local metric neighborhoods [89; 105] in the raw signal space. These ideas overlap to a considerable extent when the underlying structure is geometric,[2] but they can also differ conceptually when the structure is stochastic. In Figure 1, we illustrate the difference between a manifold structure and a stochastic co-occurrence structure. Leveraging these similarities, two unsupervised representation learning methods emerge from these ideas: manifold learning and co-occurrence statistics modeling. Interestingly, many of these ideas reach a low-rank spectral decomposition formulation or a closely related matrix factorization formulation [112; 89; 105; 27; 54; 44; 86; 46; 21].

The philosophy of manifold learning is that only local neighborhoods in the raw signal space can be trusted and that global geometry emerges by considering all local neighborhoods together — that is, "think globally, fit locally"[92]. In contrast, co-occurrence [25] statistics modeling offers a probabilistic view, which complements the manifold view as many structures cannot be modeled by continuous manifolds. A prominent example comes from natural language, where the raw data does not come from a smooth geometry. In word embedding [77; 78; 86], "Seattle" and "Dallas" might reach a similar embedding though they do not co-occur the most frequently. The underlying reason is that they share similar context patterns [68; 118]. The probabilistic and manifold points of view complement each other for understanding "similarity." With a definition of similarity, the next step is to construct a non-trivial transform such that similar things are placed closer to one another.

**How do we establish the representation transform? Through the parsimonious principles of sparsity and low-rank spectral embedding.** The general idea is that we can use sparsity to decompose and tile the data space to build a support, see Figure 2(a,b). Then, we can construct our representation transform with low-frequency functions, which assign similar values to similar points on the support, see Figure 2(c). This process is called *the sparse manifold transform* [21].

---

[1]An obvious trivial solution is to collapse every data point to a single point in the new space.

[2]If we consider that signals which temporally or spatially co-occur are related by a smooth transformation, then these three ideas are equivalent.

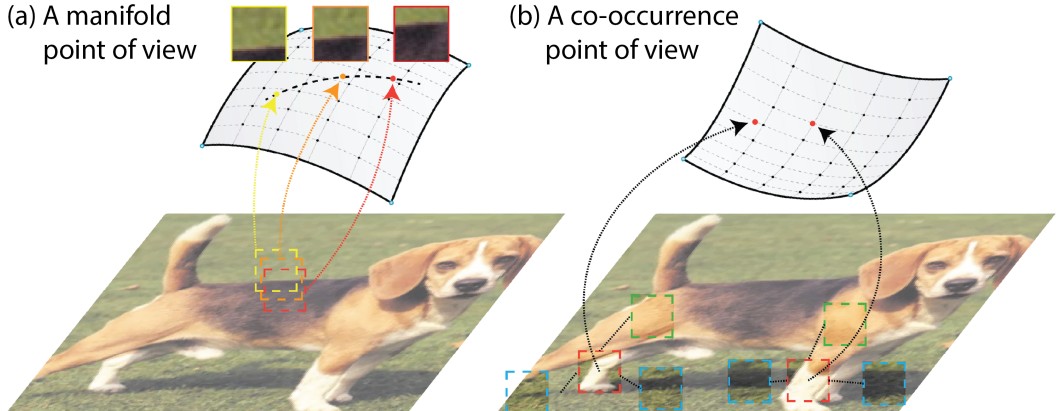

Figure 1: **Two different points of view.** (a) Manifold: There are three patches in the dashed boxes. Let's assume there is a path between them on a manifold of the image patch space; then, their relationship can be found based on either the Euclidean neighborhoods, spatial co-occurrence, or temporal co-occurrence from smooth transformation. (b) Probabilistic: For the two red boxes ("leg"), it is unclear whether there is a path between them in the patch space. However, they can still be considered to be similar under spatial co-occurrence because they appear in the same context. These two patches both co-occur with the "grass land" (blue boxes) and "elbow or body" (green boxes). Temporal co-occurrence has a similar effect.

## 2 UNSUPERVISED LEARNING WITH THE SPARSE MANIFOLD TRANSFORM

**The sparse manifold transform.** The sparse manifold transform (SMT) was proposed to unify several fundamental unsupervised learning methods: sparse coding, manifold learning, and slow feature analysis. The basic idea of SMT is: 1) first to lift the raw signal $x$ into a high-dimensional space by a non-linear sparse feature function $f$, i.e. $\vec{\alpha} = f(\vec{x})$, where locality and decomposition are respected. 2) then to linearly embed the sparse feature $\vec{\alpha}$ into a low-dimensional space, i.e., find $\vec{\beta} = P\vec{\alpha}$, such that the similarity or "trajectory linearity"[21] is respected. As we mentioned earlier, for each data point $x_i$, we can choose to use either its local neighbors in the raw signal space, its spatial co-occurrence neighbors, or its temporal co-occurrence neighbors to define similarity. Let's denote the neighbors of $\vec{x}_i$ as $\vec{x}_{n(i)}$, and denote $\vec{\alpha}_i$ as the sparse feature of $\vec{x}_i$. The optimization objective of finding similarity-preserving linear projection $P$ is the following:

$$\min_P \sum_i \sum_{j \in n(i)} \|P\vec{\alpha}_i - P\vec{\alpha}_j\|_F^2, \quad \text{s.t.} \quad P\, \mathbb{E}[\vec{\alpha}_i \vec{\alpha}_i^T] P^T = I \tag{1}$$

Example sparse feature extraction function $\vec{\alpha} = f(\cdot)$ could be sparse coding [82] and vector quantization [42], which will be further explained in Section 3.

Similarity can be considered as a first-order derivative (see Appendix L for detailed explaination). A more restrictive requirement is linearity [89], which is the second-order derivative. Then the objective is changed to a similar one:

$$\min_P \sum_i \|P\vec{\alpha}_i - \sum_{j \in n(i)} w(i,j) P\vec{\alpha}_j\|_F^2, \quad \text{s.t.} \quad P\, \mathbb{E}[\vec{\alpha}_i \vec{\alpha}_i^T] P^T = I \tag{2}$$

where $\sum_{j \in n(i)} w(i,j) = 1$. Several ways to acquire $w(\cdot, \cdot)$ are: to use locally linear interpolation, temporal linear interpolation, or spatial neighbor linear interpolation. If we write the set $\{\vec{\alpha}_i\}$ as a matrix $A$, where each column of $A$ corresponds to the sparse feature of a data point, then both Optimization 1 and Optimization 2 can be formulated as the following general formulation [21]:

$$\min_P \|PAD\|_F^2, \quad \text{s.t.} \quad PVP^T = I \tag{3}$$

where $D$ is a differential operator, $V = \frac{1}{N} AA^T$ and $N$ is the total number of data point. $D$ corresponds to a first-order derivative operator in Optimization 1 and a second-order derivative operator in Optimization 2. For Optimization 1, $D$ has M columns, where $M$ denote the number of pair

of neighbors. $D = [\mathbf{d_1}, \mathbf{d_2}, \cdots \mathbf{d_M}]$ and for $k^{th}$ pair of neighbors $x_i, x_j$, we have $\mathbf{d_{k}}_i = 1$ and $\mathbf{d_{k}}_j = -1$. All other elements of $D$ are zero. For Optimization 2, $D_{ji} = -w(i, j)$ for $j \in n(i)$, $D_{ii} = 1$, and $D_{ji} = 0$ otherwise. The solution to this generalized eigen-decomposition problem is given [109] by $P = UV^{-\frac{1}{2}}$, where $U$ is a matrix of $L$ trailing eigenvectors (i.e. eigenvectors with the smallest eigenvalues) of the matrix $Q = V^{-\frac{1}{2}} ADD^T A^T V^{-\frac{1}{2}}$. Please note that if $U$ contains all of the eigenvectors of $Q$, then the solution of SMT reduces to the whitening matrix.

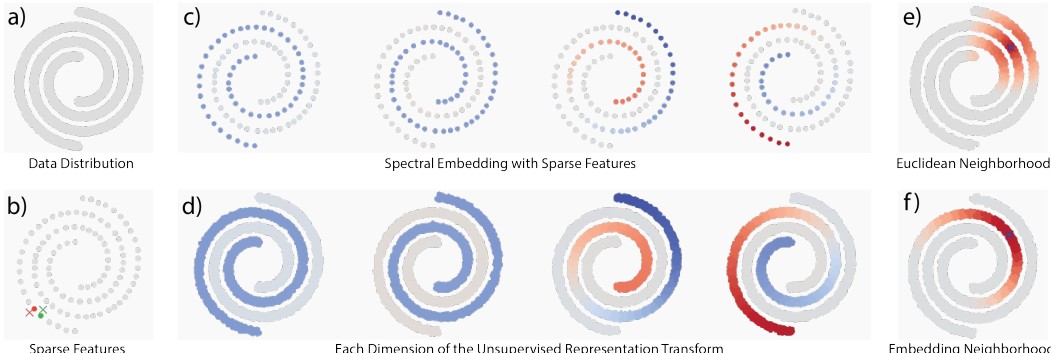

Figure 2: **Manifold disentanglement with SMT.** (a) Data distribution along two entangled 1D spiral manifolds in the 2D ambient space. (b) Sparse feature (dictionary) learned with 1-sparse sparse coding is shown as dots. They serve as a set of landmarks to tile the data manifolds. Given two points marked as red and green crosses, the 1-sparse feature function maps them to the corresponding dictionary element, marked as red and green dots. (c) Spectral embedding assigns similar values to nearby dictionary elements. Each embedding dimension is a low-frequency function defined on the sparse dictionary support. We show the first 4 embedding dimensions, i.e., the first 4 rows of $P$, $P_{1,\cdot}, P_{2,\cdot}, P_{3,\cdot}$, and $P_{4,\cdot}$. The color represents the projection values for each dictionary element. (d) The sparse manifold transform is a vector function $Pf(\cdot)$, and we plot the first four dimensions of SMT over the data distribution. The color represents the projection values for each data point. (e) We show the Euclidean neighborhood of the point marked by a blue cross in the original data space. (f) After the sparse manifold transform, we show the cosine similarity neighborhood of the point marked by the same blue cross in the representation space.

**Manifold point of view: a 2D nonlinear manifold disentanglement problem.** Let's consider a spiral manifold disentanglement problem in a 2D space shown in Figure 2(a). As it is not a linearly-separable problem, to construct a representation function to disentangle the two spiral manifolds, we need to leverage nonlinearity and give the transform function enough flexibility. Let's first use 1-sparse sparse coding to learn an overcomplete dictionary $\Phi \in \mathbb{R}^{2 \times K}$ [3], i.e. a set of $K$ landmarks [98; 27], to tile the space. This dictionary serves as the support of the representation function, as shown in Figure 2(b). This dictionary turns a data point $\vec{x}$ into a 1-sparse vector $\vec{\alpha} = f(\vec{x}) = \operatorname{argmin}_{\vec{\alpha}} \|\vec{x} - \Phi\vec{\alpha}\|$, s.t. $\|\vec{\alpha}\|_0 = 1, \|\vec{\alpha}\|_1 = 1$; $\vec{\alpha} \in \mathbb{R}^K$ is an 1-hot vector and denotes the closest dictionary element $\vec{\phi}$ to $\vec{x}$. Next, we seek a linear projection $P$ to embed the sparse vector $\vec{\alpha}$ to a lower dimensional space, $\vec{\beta} = Pf(\vec{x})$ and $\vec{\beta} \in \mathbb{R}^d$, which completes the transform. Each column of $P \in \mathbb{R}^{d \times K}$ corresponds to a $d$-dimensional vector assigned to the corresponding entry of $\alpha$. The $i^{th}$ row, $P_{i,\cdot}$, of $P$ is an embedding dimension, which should assign similar values to the nearby landmarks on the manifolds. As we mentioned earlier, there are three localities to serve as similarities, where they behave similarly in this case. With either of the three, one can define the Optimization 1 or Optimization 2. In this simple case, the different similarity definitions are almost identical, leading to a spectral solution of $P$. Let's choose a simple construction [69]: given a data point $\vec{x} \in \mathbb{R}^2$, add Gaussian noise $\epsilon_1, \epsilon_2$ to it to generate two augmented samples $\vec{x}_1 = x_1 + \epsilon_1$ and $\vec{x}_2 = x_2 + \epsilon_2$, which are marked as the red and green crosses in Figure 2(b) and are considered as similar data points. $f(\vec{x}_1)$ and $f(\vec{x}_2)$ are marked as red and green dots (landmarks). So the objective in Optimization 1 is to find $P$ such that $\mathbb{E}_{x \sim p(x), \epsilon_1, \epsilon_2 \sim \mathcal{N}(0, I)} \|Pf(x_1 + \epsilon_1) - Pf(x_2 + \epsilon_2)\|^2$ is minimized. We show the solution in Figure 2(c) with the first four embedding dimensions, $\{P_{1,\cdot}, P_{2,\cdot}, P_{3,\cdot}, P_{4,\cdot}\}$ plotted, where the color reflect their values at each dictionary element. As we can see, each embedding

---

[3] 1-sparse sparse coding is equivalent to clustering. Sampling the data as the dictionary also suffices here.

dimension, e.g. $P_{i,\cdot}$, is essentially a low-frequency function. $P_{1,\cdot}$, $P_{2,\cdot}$ are piecewise constant functions on the union of the manifolds. Essentially, either $P_{1,\cdot}$ or $P_{2,\cdot}$ will suffice to determine which manifold a point belongs to. $P_{2,\cdot}$, $P_{3,\cdot}$ are non-constant linear functions along the manifolds. A linear combination of $P_{3,\cdot}$ and $P_{4,\cdot}$ will preserve the coordinate information of the manifolds. Now with both the sparse feature function $f$ and the spectral embedding $P$, we have constructed a sparse manifold transform, i.e. $\vec{\beta} = Pf(\vec{x})$. In Figure 2(d), we show how each dimension of this transform assigns different values to each point in the original 2D space, and the points from the same manifold are placed much closer in the representation space as shown in Figure 2(f).

**Probabilistic point of view: co-occurrence modeling.** Many related variations of spectral and matrix factorization methods have been widely used to build continuous word representations. SMT formulation bears a probabilistic point of view, which complements the manifold perspective. A notable example is the word co-occurrence problem. Here we present an SMT-word embedding. Given a corpus and a token dictionary, we can treat each token as a 1-hot vector $\vec{\alpha}$. Then we can seek a linear projection $P$ such that tokens co-occur within a certain size context window are linearly projected closer. After we solve Optimization 1, each column of $P$ corresponds to a continuous embedding of an element from the token dictionary. We train SMT on WikiText-103 [76] and provide the visualization result in Appendix A. As we discussed earlier, "Seattle" and "Dallas" may reach a similar embedding though they do not co-occur the most frequently. The probabilistic view of SMT explains the phenomenon that two embedding vectors can be close in the representation space if they frequently appear in similar contexts, though they may not be close in the signal space.

**Sparsity, locality, compositionality, and hierarchy.** (a) Sparsity: In the sparse manifold transform, sparsity mainly captures the locality and compositionality, which builds a support for the spectral embedding. Spectral embedding, on the other hand, would assign similar values to similar points on the support. (b) Locality: In a low-dimensional space, e.g., the 2D manifold disentanglement case, with enough landmarks, using 1-sparse feature $f_{vq}$ to capture locality will suffice, and SMT can directly solve the problem as we showed earlier. Please note that, in this simple case, this process is not different from manifold learning [89; 105] and spectral clustering [80]. One can also use a kernel, e.g. RBF kernel or the heat kernel, to respect locality by lifting to a higher or infinite dimensional space [95; 112; 6; 102; 8]. (c) Compositionality: If the signal comes from a high-dimensional space, e.g. images, then we may need to build the representation for image patches first since even local neighborhoods can become less reliable in high-dimensional space with finite samples. If the patch space is simple enough, e.g. MNIST, using 1-sparse vector quantization as the sparse feature $f_{vq}$ would be sufficient to capture the locality in patch space. As we will show later in Section 3, the SMT with 1-sparse feature $f_{vq}$ achieves over 99.3% top-1 KNN accuracy. For more complex natural images, e.g. CIFAR or ImageNet, compositionality exists even in the patch space. So 1-sparse feature becomes less efficient and may compromise the fidelity of the representation. As we will show, the general sparse feature will achieve better performance than the 1-sparse feature in this case. (d) Hierarchy: For high-resolution images, we may need to stack several layers of transforms in order to handle the hierarchical representation [108; 35; 52].

**SMT representation for natural images.** We use SMT to build representation for image patches and aggregate patch representation of an image as the image representation. For an $S \times S$ image, we extract every $T \times T$ image patches as shown in Figure 3. Each of the image patches is whitened and L2-normalized, which resembles the early visual system processing [110; 74]. Given an image, we denote the pre-processed image patches as $\{\vec{x}_{ij}\}$, where $i$ indicates the vertical index of the patch and $j$ indicates the horizontal index. For each image patch $\{\vec{x}_{ij}\}$, we define its neighbors to be other patches within a certain pixel range from it on the same image, i.e., $n(ij) = \{ lm; |i - l| \leq d$ and $|j - m| \leq d\}$. Given a sparse feature function $f$, we would like to map these neighboring patches to be close in the representation space. If we choose $d = S - T + 1$, then all the patches from the same image are treated as neighbors to each other. Given $f$, $d$, and a set of training images, we can calculate $ADD^T A^T$, $V$, and solve Optimization 1 for $P$. [4] The representation of patch $\vec{x}_{ij}$ is $\vec{\beta}_{ij} = Pf(\vec{x}_{ij})$ with a L2 normalization. If we denote all of the patch representations from an image as tensor $B$, the image representation is $\vec{z} = \text{L2\_normalize\_pw}\,(\text{avg\_pool}(B; \text{ks}, \text{stride}))$. $\text{avg\_pool}(\cdot; \text{ks}, \text{stride})$ denotes the average pooling operator, where ks and stride are the kernel size and stride size of the pooling respectively. L2\_normalize\_pw denotes a point-wise L2 normalization. We use tensor $\vec{z}$ as a vector for an image representation. The detailed settings, e.g. $d$, ks, stride, etc.,

---

[4]While a SGD algorithm can be used[21], we use deterministic algorithm in this paper.

are left to the experiments section. In this work, we primarily use the soft-KNN classifier to evaluate the quality of representation.

**Connection to the deep latent-embedding methods.** There are important connection between the SOTA latent-embedding SSL methods[18; 5]. In particular, Optimization 1 can be considered the simplest form of VICReg [5], whereas the VC regularization is replaced by a hard constraint, and a sparse feature function replaces the deep neural encoder. See exact formulation in Appendix K. This connection is even stronger given the evidence that VICReg is essentially building a representation of image patches [22]. In this sense, SMT can be considered the simplest form of VICReg.

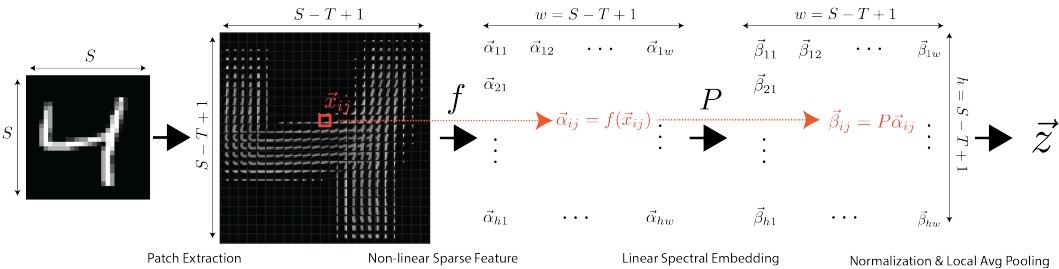

Figure 3: **SMT representation of an image and its patches.** An $S \times S$ image is first decomposed into $(S - T + 1) \times (S - T + 1)$ overlapping image patches of size $T \times T$. Each image patch is whitened and the sparse feature is calculated, $\vec{\alpha}_{ij} = f(\vec{x}_{ij})$, where $ij$ indicates the spatial index of the image patch and $\vec{x}_{ij}$ is the whitened patch. The representation of patch $\vec{x}_{ij}$ is $\vec{\beta}_{ij} = Pf(\vec{x}_{ij})$ with a L2 normalization. We then aggregate these patch representations $\vec{\beta}_{ij}$ via spatial local average pooling and L2-normalize at every spatial location to obtain the final image-level representation $\vec{z}$.

## 3 EMPIRICAL RESULTS

In this section, we first provide the benchmark results on MNIST, CIFAR-10, and CIFAR-100. Qualitative visualization is also provided for a better understanding. We also compare 1-sparse feature $f_{vq}$ and a more general sparse feature $f_{gq}$. $\vec{\alpha} = f_{vq}(\vec{x}; D)$ refers to the 1-sparse vector quantization sparse feature, where $\vec{\alpha}_i = 1$ for $i = \text{argmax}_i \|\vec{x} - D_{\cdot,i}\|_2$ and $D$ is a dictionary. We denote the SMT representation with $f_{vq}$ sparse feature as SMT-VQ. $\vec{\alpha} = f_{gq}(\vec{x}; D, t)$ refers to a more general sparse function, where each column of dictionary $D$ is an image patch and $t$ is a given threshold. For $\vec{\alpha} = f_{gq}(\vec{x}; D, t)$, the $i$th entry of $\vec{\alpha}$ is 1 if only if the cosine similarity between $\vec{x}$ and $i$th dictionary element pass the threshold, i.e., $\alpha_i = 1$ if $cos(\vec{x}, D_{\cdot,i}) \geq t$ and $\alpha_i = 0$ otherwise. We denote the SMT representation with $f_{gq}$ feature as SMT-GQ. We find $f_{gq}$ achieves similar performance as the sparse coding feature in benchmarks, and it's more efficient in experiments since it does not involve a loopy inference. For both MNIST and CIFAR, we use 6x6 image patches, i.e., $T = 6$. All the image patches are whitened, and the whitening transform is described in Appendix C. Once patch embeddings are computed, we use a spatial average pooling layer with ks = 4 and stride = 2 to aggregate patch embeddings to compute the final image-level embeddings.

**MNIST.** For MNIST, 1-sparse feature $f_{vq}$ is sufficient to build good support of the patch space. The dictionary $D \in \mathbb{R}^{36 \times K}$ is learned through 1-sparse sparse coding, which is equivalent to clustering and can be implemented with any clustering algorithm. The centroid of each cluster will serve as a dictionary element. The detailed learning procedure is left to Appendix D. The inference is equivalent to vector quantization. We chose the patch embedding dimension as 32 because further increasing the dimension does not improve the performance. In Table 1, we compare different dictionary sizes and different spatial co-occurrence ranges. We also compare the SMT embedding with the GloVe embedding. Treating each image patch as a token and the entire image patch as a sentence, we can train a GloVe word embedding model on MNIST patches. We provide a GloVe embedding with optimized settings. As we can see in Table 1, GloVe embedding works slightly worse in this case, even with a large dictionary size. Further, we can find that a small context range works slightly better for MNIST. This is probably due to the fact that the patch space of MNIST is closer to a manifold, and co-occurrence is less informative in this case. To further verify this hypothesis, we provide an ablation study on embedding dimensions in Table 2. Even if we reduce

the embedding dimension to 4 [5], the top-1 KNN accuracy is still $98.8\%$. This implies that the MNIST patch space might be close to a manifold like high-contrast natural image patches [67; 26; 12].

Table 1: **Evaluation of SMT representation on MNIST.** The table shows the evaluation result of SMT embedding and GloVe embedding using a KNN classifier. We compare different dictionary size: 16384 vs. 65536 and context range: 3-Pixel ($d = 3$) vs. Whole Image ($d = 28 - 6 + 1$). We use 32 as the embedding dimension. As shown in the table, GloVe embedding works slightly worse than SMT embedding.

| Co-Occurrence Context Range | SMT-VQ (16384) | SMT-VQ (65536) | SMT-VQ (GloVe, 100K) |
|---|---|---|---|
| Whole Image | 99.0% | 98.9% | 98.8% |
| 3 Pixels | 99.2% | 99.3% | 99.0% |

Table 2: **Low dimensionality of MNIST patch space.** The table shows the effect of the embedding dimension of SMT on the KNN evaluation accuracy. With only a 4-dimensional embedding for each patch, SMT can still achieve 98.8% KNN accuracy. The quick accuracy drop at 3-dimensional embedding space may imply that the MNIST patch space can be approximately viewed as a 3-manifold since the 4-dimensional patch embedding is L2-normalized and on a 3-sphere.

| | Emb Dim=2 | Emb Dim=3 | Emb Dim=4 | Emb Dim=8 | Emb Dim=16 | Emb Dim=32 |
|---|---|---|---|---|---|---|
| KNN Acc | 97.1% | 98.3% | 98.8% | 99.0% | 99.1% | 99.2% |

**CIFAR-10 and CIFAR-100.** For the CIFAR benchmarks, we compare both $f_{vq}(\cdot; D)$ and $f_{gq}(\cdot; D, t)$. The dictionary $D$ used in $f_{vq}$ is learned through 1-sparse sparse coding, and the dictionary $D$ used in $f_{gq}$ is just randomly sampled from all image patches for efficiency purposes. Since $f_{vq}$ with color patches underperforms that with grayscale patches, we choose to experiment mostly with $f_{gq}$ on CIFAR benchmarks. For $f_{gq}$, $t$ is set to $0.3$ with color patches and $0.45$ with grayscale patches. We choose the patch embedding dimension as 384. Further, with color patches, dropping the first 16 embedding dimensions of $P$ improves the accuracy by about 1%. We compare the performance of SMT with SOTA SSL models, specifically, SimCLR and VICReg with a ResNet 18 backbone. We train these SSL models using random resized crop and random horizontal flip as default data augmentations. For a fair comparison, SMT training uses both the original version and the horizontally flipped version of the training set. We also experiment with different color augmentation schemes. For "Grayscale Image Only", we train both SMT and SSL models using only grayscale images. For "Original + Grayscale", we train SMT on both color images and grayscale images, then concatenates the embedding of each color image with the embedding of its grayscale version. We train SSL models with additional "random grayscale" color augmentations. With a two-layer deterministic construction using only image patches, SMT significantly outperforms SimCLR and VICReg, with no color augmentation. With "Grayscale Image Only" color augmentation, SMT-GQ achieves similar performance as SimCLR and VICReg. With "Original + Grayscale" augmentation, SMT-GQ performs on par with SimCLR and VICReg without colorjitter and several additional color augmentations. With full data augmentation, SimCLR and VICReg outperform SMT-GQ. However, we did not find a straightforward way to add these color augmentations to SMT-GQ. We can observe the same trend on CIFAR-100. Interestingly, SMT-GQ always significantly outperforms SimCLR and VICReg before we add color jittering and additional data augmentations. We provide further ablation studies in Appendix. Appendix G and Appendix H study how context range and the number of training samples affect the model performance. Appendix I studies different evaluation schemes (KNN vs. linear probing). Appendix E shows that the performance of SMT will significantly degrade if we replace spectral embedding with other dimension reduction algorithms that do not respect locality.

**Visualization.** For a better understanding, we provide a visualization of the patch embedding space with the 1-sparse feature in Figure 4. Given a dictionary $D$, each dictionary element is an image patch in the whitened space. For a dictionary element shown in the red box, we can visualize its nearest neighbors, measured by cosine similarity, in the embedding space. All the image patches are unwhitened before visualization, where the detail is left to the Appendix C. In Figure 4, we can see that the nearest neighbors are all semantically similar. a) is a slightly curved stroke of a digit in MNIST. b) shows the patch of the center of a digit "4". c) shows a part of a "wheel" of a car

---

[5]As the embedding is further L2 normalized, this is a 3-dimensional space.

Table 3: **Evaluation of SMT and SSL pretrained model on CIFAR-10.** The table shows the evaluation result of SMT embedding and SSL pretrained models using KNN classifiers. We evaluated these models using various color augmentation schemes. The details of how we add color augmentation to each model are explained in section 3. We tried two sparse features for SMT: $f_{vq}$ and $f_{gq}$. For $f_{gq}$, we tried two dictionary sizes: 8192 and 65536. Both SSL models use ResNet-18 as a backbone and are optimized for 1000 epochs through backpropagation; SMT uses a 2-layer architecture and is analytically solved, which can be considered as 1 epoch.

| Color Augmentation | SMT-VQ (100K) | SMT-GQ (8192) | SMT-GQ (65536) | SimCLR (ResNet18) | VICReg (ResNet18) |
|---|---|---|---|---|---|
| Original Image | — | 79.2% | **81.1%** | 68.3% | 70.2% |
| Grayscale Image Only | 78.4% | 77.5% | 78.9% | 80.6% | **81.3%** |
| Original + Grayscale | — | 81.4% | 83.2% | **85.7%** | 83.7% |
| Full (ColorJitter etc.) | — | — | — | 90.1% | **91.1%** |

Table 4: **Evaluation of SMT and SSL pretrained model on CIFAR-100.** The table shows the performance of SMT embedding and two other SSL pretraining methods on the CIFAR-100 dataset. The data augmentation and different SMT variations we used in this experiment are the same as those for evaluating CIFAR-10 dataset in Table 3. For CIFAR-100 dataset, without full color augmentation, SMT always outperforms the two SSL pretraining methods.

| Color Augmentation | SMT-VQ (100K) | SMT-GQ (8192) | SMT-GQ (65536) | SimCLR (ResNet18) | VICReg (ResNet18) |
|---|---|---|---|---|---|
| Original Image | — | 50.8% | **53.2%** | 32.4% | 32.6% |
| Grayscale Image | 46.6% | 45.8% | **48.9%** | 43.0% | 43.9% |
| Original + Grayscale | — | 53.7% | **57.0%** | 48.9% | 46.0% |
| Full (ColorJitter etc.) | — | — | — | 63.7% | **65.4%** |

in CIFAR. d) shows a part of "animal face". These results are similar to the 2D results shown in Figure 2. Overall, the representation transform brings similar points in the data space closer in the representation space. Additional visualization for SMT on CIFAR-10 dataset is in Appendix M.

## 4 MORE RELATED WORKS

There are several intertwined quests closely related to this work. Here, we touch them briefly.

**The state-of-the-art unsupervised methods.** Deep learning [63; 66] is the engine behind the recent revolution of AI. Given the right craftsmanship, deep neural networks are powerful optimization tools to allow us express various engineering objectives conveniently. It has also revolutionized the unsupervised visual representation learning in the past few years [113; 47; 18; 5]. Understanding the engineering advancements can help us gain deeper theoretical insights and many efforts have been made [85; 1; 70; 2; 94] towards this goal.

**To model the patterns of the signal spaces.** A central challenge of modeling the signals is the formalization of a small set of ideas for constructing the representations of the patterns[43; 79]. Wavelet transform[72; 24] was formulated to handle the sparsity in the natural signals. Steerable and shiftable filters[38; 101; 99] were introduced to handle the discontinuity of wavelet filters with respect to the natural transformations. Basis pursuit[17; 16] finds decomposition of a signal by a predefined dictionary, sparse coding[81; 82], and ICA[7; 56] were proposed to learn the filters or dictionary from data. Before the sparse manifold transform[21], many important works have paved the way and introduced topology and transformations between the dictionary elements[57; 58; 11]. There are also several important works studying the natural image patch space with non-parametric methods and topological tools [67; 26; 12; 48]. Manifold learning[89; 105; 44] and spectral clustering [33; 37; 95; 75; 80; 104; 93] was proposed to model the geometric structure in the signal space. The geometric signal processing is also an increasingly important topic[96].

**Modeling of biological visual system.** Exploring the geometric or statistical structure of natural signals are also strongly motivated by computational modeling of biological visual system. Many properties of the visual system reflect the structure of natural images [100; 59]. An early success of this quest was that the dictionary elements learned in an unsupervised fashion closely resemble the receptive fields of simple cells in the V1 cortex[81; 82]. In order to further understand the visual cortex, we need to develop theories that can help to elucidate how the cortex performs the computationally challenging problems of vision [83]. One important proposal is that the visual cortex performs hierarchical manifold disentanglement [31; 32]. The sparse manifold transform

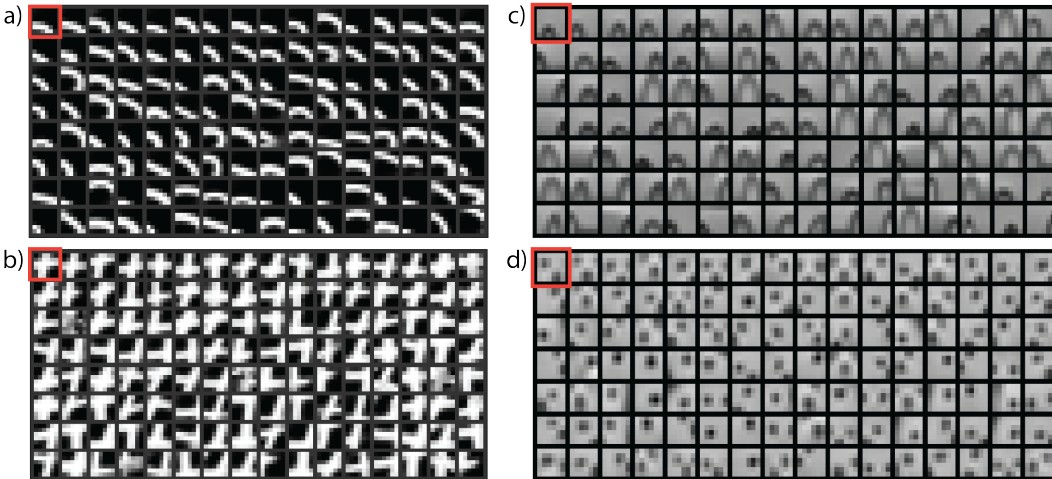

Figure 4: **Visualization of dictionary elements' neighborhoods in the SMT embedding space.** We plot 4 different dictionary element and their top nearest neighbors in four subfigures, respectively. For each subfigure, the dictionary element is shown in the red box. Its neighboring patches in the representation space are ordered by their cosine similarity to the dictionary element, where the image patches in the top left have highest cosine similarity, and image patches on the bottom right have lowest cosine similarity. This figure is qualitatively similar to Figure 2(f) except it is in a high-dimensional space and the semantic meaning is more complex.

was formulated to solve this problem, which is supported by a recent perceptual and neuroscience evidence [49; 50]. There are also several related seminal works on video linearization [112; 41; 40].

**To build representation models from the first principles.** The now classical models before deep learning use to be relatively simpler [64; 108; 87; 60; 111; 115; 61], where sparsity was a widely adopted principle. While the gap between the SOTA deep learning models and the simple models is large, it might be smaller than many expect [23; 88; 10; 106]. The quest to build "white-box" models from the first principles [14; 13; 71] for unsupervised representation remains an open problem.

## 5 DISCUSSION

In this paper, we utilize the classical unsupervised learning principles of sparsity and low-rank spectral embedding to build a minimalistic unsupervised representation with the sparse manifold transform [21]. The constructed "white-box" model achieves non-trivial performance on several benchmarks. Along the way, we address fundamental questions and provide two complementary points of view of the proposed model: manifold disentanglement and co-occurrence modeling. Further, the proposed method has close connections with VICReg [5] and several other SOTA latent-embedding SSL models [18; 117; 39]. However, there are several important limits to this work, which point to potential future directions. 1) We need to scale the method to larger datasets, such as ImageNet. 2) While a two-layer network achieves non-trivial performance, we need to further construct white-box hierarchical models to understand how a hierarchical representation is formed [51; 53]. 3) While spectral embedding is context-independent [86; 118], there is clear benefit in building contextualized embedding [29; 91; 52; 36; 116; 73]. Building a minimalistic unsupervised representation based on contextualized embedding is an attractive topic for additional research. 4) Although we only focus on sparse feature in this paper, other features should also be explored. We can generalize feature extraction function $f$ in SMT to an arbitrary function as long as it can capture the locality and structures in the signal space. For example, if we choose $f$ as a quadratic expansion function, Optimization 1 would reduce to slow feature analysis [112].

The evidence shows that there considerable promise in building minimalistic white-box unsupervised representation learning models. Further reducing the gap with SOTA methods and understanding the mechanism behind this reduction can lead us towards a theory. Minimalistic engineering models are easier to analyze and help us build a theory to identify optimality and efficiency, and a testable theory can also facilitate modeling biological learning systems. This work takes a step towards these goals.

## ACKNOWLEDGMENTS

We thank Zengyi Li, Jamie Simon, Sam Buchanan, Juexiao Zhang, Christian Shewmake, Jiachen Zhu, Domas Buracas, and Shengbang Tong for participating in several intriguing discussions during the preparation of this manuscript. We also thank Fritz Sommer and Eero Simoncelli for providing valuable feedback. We also thank our anonymous reviewers for the helpful suggestions and insightful questions, which helped us improve the manuscript considerably.

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

# Appendix

## A   PROBABILISTIC POINT OF VIEW EXAMPLE: SMT WORD EMBEDDING

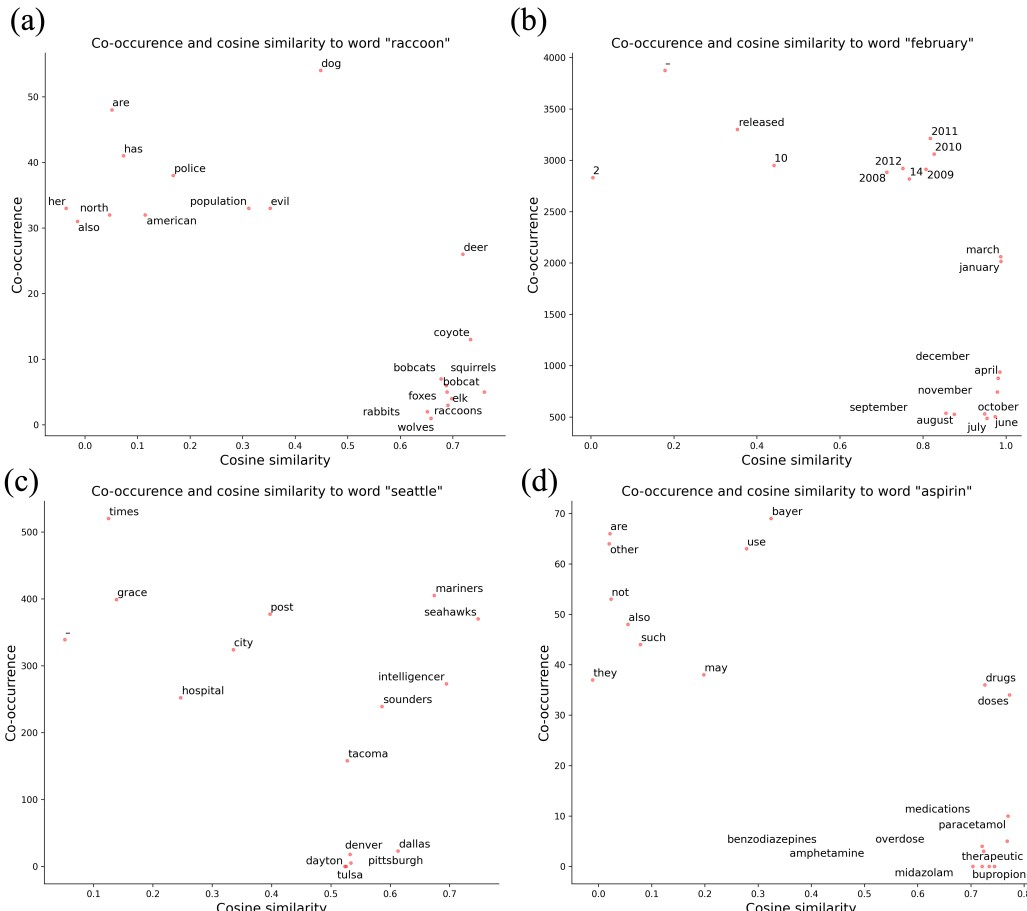

Figure 5: **Visualization of SMT word embedding.** For each word "raccoon", "February", "Seattle" and "aspirin", we plot the top 10 words that are closest to it in the embedding space, and we show the top 10 words that co-occur with it the most. Two words are close if the cosine similarity between their SMT embedding is high. The co-occurrence is calculated based on the entire corpus. We count two words that co-occurred once if they appear in the same context window. The co-occurrence is plotted against the cosine similarity.

We use SMT to build word embedding for the vocabulary in WikiText-103 to illustrate the probabilistic view of SMT. Unlike in Figure 2, where the neighbor of each data point $x$ can be interpreted as its Euclidean neighborhood, there is no such distance function in the natural language. Nevertheless, we could still define neighborhood for each word. We define the neighbor of each word as other words that co-occur with it in a given context.

Like the manifold learning view, where the global geometry emerges despite only local neighborhoods being used during training, such a phenomenon is also observed from the probabilistic point of view. To visualize this phenomenon from a probabilistic point of view, we measure the co-occurrence of each pair of words and the cosine similarity between them in their SMT embedding space. In figure 5 (a), we show the top 10 words that are closest to "raccoon" and the top 10 words that co-occur the most with "raccoon". We can see words like "American," and "population" co-occur with "raccoon" a lot, but neither "American" nor "population" are close to "raccoon" in the embedding space. Instead, words that are semantically related to "raccoon" are close to "raccoon"

in the embedding space, like "squirrels", "rabbits" and "wolves", even though they rarely co-occur with "raccoon." The underlying reason is that these words share the same context patterns as "raccoon". Although words like "American" and "population" can co-occur with "raccoon" in phrases like "American raccoon" and "raccoon's population," these words also co-occur with other words in very different contexts. Thus, they could not be embedded to be close "raccoon."

We also illustrate three more examples in figure 5. In (b), we see words that occur with "February" are other months like "July," "June," and "September." In (c), words co-occur with "Seattle" the most are other large cities like "Dallas", "Denver" and "Pittsburgh", even though they rarely occur with "Seattle." Finally, in (d), the words co-occur with "aspirin" the most are names of other medicines.

## B    DETAILS FOR TRAINING WORD EMBEDDING USING SMT

**Dataset.** We use WikiText-103 corpus [76] to train SMT word embedding, which contains around 103 million tokens. We use "Basic English Tokenizer" from torchtext, which uses minimal text normalization for the tokenizer. We tokenize and lowercase the WikiText-103 corpus, building a vocabulary using the top 33,474 words with the highest frequency. Notice that this dataset is significantly smaller than the Wikipedia dump used to train GLoVe in the original paper [86]. Wikipedia is also highly biased toward technical subjects, which is reflected in our word embedding.

**SMT training.** First, we choose the one-hot encoding as the sparse extraction function $f$. Each token in the text corpus is considered a data point $\vec{x_i}$; it is first transformed into a one-hot vector by $f$, then encoded to be close to its neighbor via spectral embedding. In order to perform spectral embedding, we need to define the neighbor(context) for each word in Optimization 1. In this case, each word's neighbor(context) is its previous four words and the next four words. We can then calculate $ADD^T A^T$, $V$, and solve Optimization 1 for $P$. The representation of each word in the vocabulary is simply each column of P. This is because $f$ is one-hot encoding, for each word $\vec{x_i}$, we have $\vec{\beta_i} = Pf(\vec{x_i}) = P\mathbf{e_i} = \vec{P_i}$.

## C    WHITENING OF THE IMAGE PATCHES

We consider all the square image patches with size $T \times T \times c$, where $T$ is patch size, and $c$ is channel size, as flattened vectors of dimension $T^2 c$. We first remove the mean from each patch. Instead of calculating the global mean like in the normal whitening procedure, we calculate the contextual mean. Let $x_i$ denote a image patch and let $N_i = \{x_j : j \in n(i)\}$ denote the neighboring patch for $x_i$. We can calculate the contextual mean of $x_i$ as $\mu_i = \frac{1}{||N_i||} \sum_{j \in N_i} x_j$. Let $\bar{x_i}$ denote the mean removed image patch, then $\bar{x_i} = x_i - \mu_i$. We could also simply calculate the global mean $\mu$ and use it to center all the image patches. However, we found using contextual mean improve performance.

We apply the whitening operator to the mean-removed image patches. Let $\{\vec{x_i}\}$ denote the pre-processed image patches. Then we have

$$\vec{x_i} = (\lambda I + \Sigma)^{-\frac{1}{2}} \bar{x_i}$$

To "unwhiten" the patch for visualization, we multiply the whitened patch with the inverse of whiten operator: $(\lambda I + \Sigma)^{\frac{1}{2}}$.

## D    1-SPARSE DICTIONARY LEARNING

Given $\vec{x_i} \in \mathbb{R}^d$ to be a data vector, and dictionary (codebook) $\Phi \in \mathbb{R}^{d \times K}$, where $d$ is the dimension of the data and $K$ is the number of dictionary elements, 1-Sparse Dictionary Learning, also known as Vector Quantization (VQ problem [42]), can be formulated as:

$$\min_{\vec{\alpha_i}, \Phi} \sum_i ||\vec{x_i} - \Phi\vec{\alpha_i}||_2, \tag{4}$$

$$s.t. \quad ||\vec{\alpha_i}||_0 = 1, ||\vec{\alpha_i}||_1 = 1, \vec{\alpha_i} \succeq 0.$$

This is an NP-hard problem, but there are efficient heuristic algorithms to approximate the solution. The most well-known one is K-mean clustering, which can be interpreted as an expectation-maximization (EM) algorithm for mixtures of Gaussian distributions. This is the algorithm we used

to solve 1-sparse dictionary learning when we use the 1-sparse feature $f_{vq}$ in SMT. The exact algorithm we used to solve the 1-sparse dictionary learning problem is in Algorithm 1. Notice that when data is uniformly distributed on the domain, even a dictionary consisting of uniformly sampled data point serve as a good enough approximation. This is the case for the toy example in Figure 2. In this case, the dictionary $\Phi$ can be predefined instead of learned. For MNIST and CIFAR-10 data, they are both assumed to be low dimensional data manifolds in high dimensional space. Thus, we cannot apply k-mean directly to the raw signal due to the high dimensionality and sparsity of the data. One way to alleviate this problem is to map the data to a unit sphere before applying k-mean. This method is called spherical k-mean. More detailed motivations and advantages of this approach are explained in the original paper [30]. The exact algorithm is also described in Algorithm 1.

---

**Algorithm 1:** Online 1-sparse dictionary learning heuristic (K-mean clustering)

**Data:** Data matrix $X = [\vec{x}_1, \cdots, \vec{x}_N]$, $\eta$ is learning rate.

**Result:** Dictionary (centroid) $\Phi = [\vec{\phi}_1, \cdots, \vec{\phi}_K]$, where each column is a dictionary element
 Sparse code (label) $A = [\vec{\alpha}_1, \cdots, \vec{\alpha}_N]$ for each data point.

1 initialization: $\Phi \leftarrow 0, A \leftarrow 0$ ;
2 **if** *spherical* **then**
3     **for** $i \in 1, \cdots, n$ **do**
4        $\vec{x}_i = \frac{\vec{x}_i}{||\vec{x}_i||_2}$
5     **end**
6 **end**
7 **while** *not $\phi$ has not converge* **do**
8     *Find the sparse code for every data point*;
9     **for** $i \in 1, \cdots, n$ **do**
10        $\vec{\alpha}_i \leftarrow \vec{e}_{j^*}$    $j^* = \arg\max_j ||\vec{\phi}_j - \vec{x}_i||_2$
11     **end**
12     *Dictionary update*;
13     $\Phi \leftarrow \Phi + \frac{\eta}{h} \odot \sum_i (\vec{x}_i - \Phi^T \vec{\alpha}_i), \quad s.t. \quad \vec{h}_j = (\sum_i \vec{\alpha}_i)_j$;
14     **if** *spherical* **then**
15        **for** $j \in 1, \cdots, K$ **do**
16           $\vec{\phi}_j = \frac{\vec{\phi}_j}{||\vec{\phi}_j||_2}$
17        **end**
18     **end**
19 **end**

---

# E    ABLATION STUDY: FEATURE DIMENSION AND EMBEDDING DIMENSION

Since SMT is a white box model, we can analyze how each component of SMT contributes to unsupervised learning. Specifically, we break down SMT into two components: dimensionality expansion using sparse features and dimensionality reduction through spectral embedding. We first show that a sparse feature by itself is important in building a good representation. As shown in Figure 6, we're able to achieve 71 % evaluation accuracy using only general sparse features $f_{gq}$ on the CIFAR-10 dataset. Please refer to Section 3 for the definition of general sparse features $f_{gq}$. Combining sparse features with whitening will further boost the performance to 75 %. However, the performance does not increase as the number of features increases if we're using only sparse features and whitening. In contrast, if we combine sparse features and spectral embedding, the performance increases as the number of features increases. This result implies that spectral embedding is also a crucial element in obtaining a good representation.

To further show the necessity of spectral embedding, we did an experiment to replace spectral embedding with PCA. PCA is a dimension reduction algorithm but does not enforce locality. In Figure 6(b), we see that if we're using PCA as dimension reduction, the performance decreases monotonically as we reduce the sparse feature to a lower dimension, which implies it is throwing away useful information. On the other hand, for SMT, the performance first increases and then decreases,

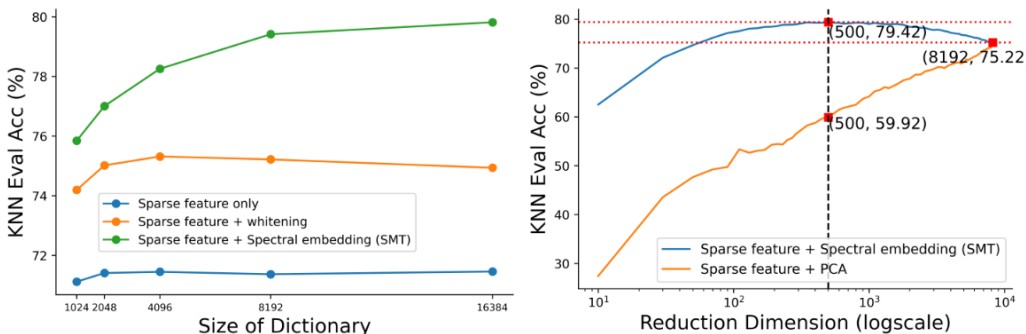

Figure 6: **Ablation Studies.** (a) Evaluation accuracy as a function of dictionary sizes on CIFAR-10 dataset. We tried three representations for ablation study purpose. (b) Evaluation accuracy as a function of reduction dimension on CIFAR-10 dataset. Reduction dimension corresponds to number of principle components in PCA and embedding dimension in SMT. We use general sparse feature $f_{gq}$ with threshold set to be 0.3, a dictionary of size 8192 and no color augmentation in this experiment.

as we increase the reduction dimension. The performance of SMT peaks when we use 500 as the reduction dimension, which is much higher than PCA (79.4% vs 59.9%) with 500 components and even higher than PCA with all principle components (79.4% vs 75.2%).

## F    Hyperparameter for pretraining deep SSL model

For all the benchmark results on the deep SSL model, we follow the pretraining procedure used in solo-learn. Each method is pretrained for 1000 epochs. We list the hyperparameter we used to pretrain each model. For both methods, we use resnet-18 as the backbone, and a batch size of 256, LARS optimizer, with 10 epochs of warmup follows by a cosine decay. For the SimCLR model, we set the learning rate to be 0.4 and a weight decay of 1e-5. We also set the temperature to 0.2, the dimension for the projection output layer to 256, and the dimension of the projection hidden layer to 2048. For VICReg, we set the learning rate to 0.3 and a weight decay of 1e-4. We also set the temperature to 0.2, and set the dimension for the projection output and hidden layer to 2048. We set the weight for similarity loss and variance loss to 25, and the weight for covariance loss to 1.

We use the transform module in pytorchvision to implement data augmentations. By default, we use "RandomResizedCrop" with min-scale equal to 0.08 and max-scale equal to 1.0. We also use "RandomHorizontalFlip" and use 0.5 as the flip probability. For "Grayscale Image Only", we apply the "RandomGrayscale" transform with the grayscale probability to 1. For "Original + Grayscale," we set the grayscale probability to 0.5. For "Full" color data augmentation, we add "ColorJitter." For SimCLR, we use brightness = 0.8, contrast= 0.8, saturation=0.8, and hue=0.2 as its "ColorJitter" parameter. For VICReg, we use brightness = 0.4, contrast= 0.4, saturation=0.2, hue=0.1 as its "ColorJitter" parameter. We also add additional "Solarization" transform with probability 0.1 for training VICReg.

## G    Ablation study: context range

Given an image patch $\{\vec{x}_{ij}\}$, we define its neighbors to be other patches within a specific pixel range from it on the same image, i.e., $n(ij) = \{ lm;\ |i - l| \leq d \text{ and } |j - m| \leq d\}$. We use $d$ to denote the "context range". We showed the impact of context range on MNIST in table 1. The performance of SMT decreases as we increase the context length. We also studied the effect of context length on the CIFAR-10 dataset and showed the result in table 5. Unlike MNIST, the performance on CIFAR-10 dataset increases as the context window increases. This once again confirms that both the manifold view and the probabilistic co-occurrence view are necessary to understand SMT. For CIFAR-10, the images have lots of high-level structures. The key to building a good representation is the ability to model the co-occurrence between different image patches. For example, in an image of an animal,

Table 5: **Ablation study on the context range used in SMT on CIFAR-10 dataset.** We use general sparse feature $f_{gq}$ with threshold set to be 0.3, a dictionary of size 8192 and no color augmentation in this experiment.

| Context Range ($d$) | 4 | 6 | 12 | 24 | 32 |
|---|---|---|---|---|---|
| KNN Accuracy | 78.0 % | 78.5 % | 78.5 % | 79.0 % | 79.4% |

Table 6: **Ablation study on the number of training examples used in SMT and SimCLR on CIFAR-10 dataset.** We use general sparse feature $f_{gq}$ with threshold set to be 0.3, a dictionary of size 8192 and no color augmentation in this experiment. We evaluate both method using soft KNN classifier.

| number of training examples | 1000 | 2000 | 4000 | 8000 | 16000 | 32000 | 50000 |
|---|---|---|---|---|---|---|---|
| SMT-GQ (8192, no color aug) | 58.7 % | 63.7 % | 67.2 % | 70.7 % | 74.0 % | 77.4 % | 79.4 % |
| SimCLR (ResNet18, no color aug) | 28.9 % | 33.9 % | 37.8 % | 41.3 % | 50.1 % | 60.4 % | 68.3 % |

the animal's "leg" and "head" might lie in two image patches far apart. But it is very important to capture the fact that "leg" and "head" co-occur, because this is a unique trait for animals, which requires the context range to be large. This also requires the probabilistic co-occurrence view. Many image patches can be neighbors (co-occur), but they cannot all be encoded to the same point. Instead, if we record the co-occurrence, patches that co-occur more often should be encoded closer than to each other. On the other hand, MNIST is closer to a manifold, so co-occurrence is less informative in this case. Modeling co-occurrence between image patches might result in overfitting. In order to form a good representation, we need to use a small context range to capture slow features. This is a lot like the formulation in the original sparse manifold paper, where they use consecutive video frames to define context, which is essentially a context range of only 2 pixels.

## H  ABLATION STUDY: NUMBER OF TRAINING EXAMPLES

We experiment with the impact of a number of training examples on performance. The experiment is done on the CIFAR10 dataset. The result is shown in table 6. The result shows SimCLR experienced a much more significant performance drop compared to SMT when training with a limited amount of data. The significant performance drop of the deep SSL method is likely due to the well-known overfitting problem of complicated models like neural networks. This shows the advantages of SMT as a simple white-box model.

## I  ABLATION STUDY: LINEAR PROBING VS KNN FOR EVALUATION

We experiment with using linear probing for evaluation instead of the KNN classifier. We also experiment with the comparison between linear probing and KNN classifiers with a different number of embedding dimensions. As shown in the table 7, with optimal embedding dimension, linear probing accuracy is higher than KNN accuracy (81.0% vs 79.4% ). This implies that the linear layer learned in SMT is not optimal for image classification, but it is still a pretty good representation for image classification purposes. Most useful information for image classification is captured during this unsupervised dimension reduction process. This explains why the optimal embedding dimension is much higher (3200) when using a linear classifier compared to when using a KNN classifier (500).

## J  SOFT KNN EVALUATION

Classic K-nearest-neighbor classifier (hard KNN): for each data point $x$, which is labeled with $k$ class, let $y, y_j$ denote the label of $x, x_j$. Then let $\hat{z(x)} = \frac{1}{|n(x)|} \sum_{x_j \in n(x)} OneHot(y_j)$, which counts the average number of time $x$'s neighbors belong to each class. And $q(x) = softmax(z(x))$,

Table 7: **Comparing the performance of using linear probing or KNN classifier for evaluation on CIFAR-10.** We use general sparse feature $f_{gq}$ with threshold set to be 0.3, a dictionary of size 8192 and no color augmentation in this experiment.

| Embedding dimension | 50 | 110 | 210 | 350 | 500 | 800 | 1600 | 3200 | 6400 | 8192 |
|---|---|---|---|---|---|---|---|---|---|---|
| KNN Accuracy | 74.7 % | 77.6 % | 78.6 % | 79.3 % | **79.4 %** | 79.3 % | 79.0 % | 77.9 % | 76.1 % | 75.2 % |
| Linear Probing Accuracy | 69.0 % | 74.5 % | 77.0 % | 78.6 % | 79.7 % | 80.2 % | 80.8 % | **81.0 %** | 80.5 % | 80.4 % |

i.e. $q(x)_k = \frac{exp(\frac{z(x)_k}{T})}{\sum_l exp(\frac{z(x)_l}{T})}$. This quantity is the estimated conditional probability that the data point $x$ belong to $k$th class, given temperature $T$, i.e. $\mathbb{P}(y = k|x)$. Finally, we simply assign the class with largest conditional probability as the predicted class label i.e. $\hat{y}(x) = \arg\max_k q(x)_k$.

Soft K-nearest-neighbor classifier (soft KNN): for a given data point $x$, instead of using only $x$'s neighbors' class label to estimate the conditional probability, we also we take into account of the cosine similarity between $x$ to its neighbors. Instead of having $\hat{z}(x) = \frac{1}{|n(x)|} \sum_{x_j \in n(x)} OneHot(y_j)$, we have $\hat{z}(x) = \frac{1}{|n(x)|} \sum_{x_j \in n(x)} OneHot(y_j)cos(x_j, x)$. The rest of the algorithm is exactly the same as hard KNN. In all the experiment, we set $K = 30$ and $T = 0.03$ for evaluation.

## K  CONNECTION TO DEEP LATENT-EMBEDDING SSL METHOD

The general formulation of SMT, as stated in Section 3, is:

$$\min_P \|PAD\|_F^2, \quad \text{s.t.} \quad PVP^T = I \tag{5}$$

where $A$ denotes the sparse representation of the data.

Feature extraction: As discussed in Section 2, sparse features are used to capture locality and compositionality. There's also other feature that might be used to serve the same purpose. We can abstract such feature extractor as a function $f : \mathcal{X} \to \mathcal{A}$.

Projection: As discussed in Section 2, we assume the sparse feature of the data is a low-dimensional manifold that lies in a high-dimensional space. Thus, we seek an optimal map that projects the sparse representation to the low dimensional space, such that points that are close in feature space should also be close after the projection. In our setup, we use a linear function to approximate this optimal projection function, but the projection function does not have to be linear in general. We can abstract this projection function as a function $g : \mathcal{A} \to \mathcal{Z}$.

Let $Z = g(f(X))$, then SMT objective can be generalized to the following form:

$$\min_{f,g} \|ZD\|_F^2, \quad \text{s.t.} \quad ZZ^T = I$$

Consider the objective for VICReg:

$$L_{vic} = \alpha \sum_{k=1}^d max(0, 1 - \sqrt{(Cov(Z)_{kk})}) + \beta \sum_{i=1, j \neq k} Cov(Z)_{kj}^2 + \frac{\gamma}{N} \sum_i \sum_{j \in n(i)} |z_i - z_j|_2^2$$

where $Z = g(f(X))$, and both feature extractor $f$ and projector $g$ are parametrized by neural networks. Like discussed in [4], we use a friendlier variance loss: replacing the hinge loss at 1 as square loss. We also assume $\alpha = \beta = \gamma = 1$ for simplicity. Then we have:

$$\begin{aligned} L_{vic} &= \sum_{k=1}^d (1 - \sqrt{(Cov(Z)_{kk})}^2 + \sum_{i=1, j \neq k} Cov(Z)_{kj}^2 + \frac{1}{N} \sum_i \sum_{j \in n(i)} |z_i - z_j|_2^2 \\ &= \|ZZ^T - I\|_F^2 + \frac{1}{N} \sum_i \sum_{j \in n(i)} \|z_i - z_j\|_2^2 \end{aligned} \tag{6}$$

Thus,

$$\min_{f,g} L_{vic} = \min_{f,g} \sum_i \sum_{j \in n(i)} \|z_i - z_j\|_2^2, \quad \text{s.t.} \quad ZZ^T = I$$

This is the same objective as Eq 1. We can construct $D$ as the same first order derative operator as the one described in Section 2, then we have:

$$\min_{f,g} \|ZD\|_F^2, \quad \text{s.t.} \quad ZZ^T = I$$

With generalized features and projects, we can formulate VICReg as a sparse manifold transform problem. In VICReg, there is no close-form solution since both parameters $f$ and $g$ are parametrized by neural networks. They are optimized with a stochastic gradient descent-based algorithm. As discussed in [4], most SSL models can be reduced into the same form, which means they can all be formulated into SMT objectives.

## L  INTERPRETING SIMILARITY LOSS AS FIRST-ORDER DIRECTIONAL DERIVATIVE

Similarity can be considered as a first-order directional derivative of the representation transform w.r.t. the similarity neighborhoods. A representation transform $g$ maps a raw signal vector $\vec{x}$ into a representation embedding $\vec{\beta}$, i.e., $g : \mathbb{R}^m \to \mathbb{R}^d$. In SMT, $g = Pf(\cdot)$, where $f$ is a sparse feature function.

The definition of directional derivative is: $D_{\hat{u}}g(\alpha) = \lim_{h \to 0} \frac{g(\alpha + h\hat{u}) - g(\alpha)}{h}$ ,where $\hat{u}$ is the direction we want to take the derivative and $h$ is the infinitesimal change. In a discrete space, we represent the dataset as a graph. Each data point is a node that connects to its neighbors. As we discussed at the beginning of the paper, "similarity" neighborhoods can be defined by 1) temporal co-occurrence, 2) spatial co-occurrence, 3) local metric neighborhoods in the raw signal space. The direction between two neighboring raw signal points $x_i$ and $x_j$ is defined as their direction in the sparse feature space, i.e., $f(x_j) - f(x_i)$. Unlike in the real coordinate space, where we can take infinite direct directions at each location, with the discrete setting, we can only take a finite number of directions at each data point (the same number as the number of its neighbors). Moreover, since everything is discrete, the infinitesimal change $h = 1$.

For example, we want to take the directional derivative at point $x_i$ and in the direction towards its neighbor $x_j$. Then we have $\hat{u} = f(x_j) - f(x_i)$, and

$$\begin{aligned}
D_{\hat{u}}g(x_i) &= D_{\hat{u}}g(f(x_i)) \\
&= \lim_{h \to 0} \frac{g(f(x_i) + h\hat{u}) - g(f(x_i))}{h} \\
&= g(f(x_i) + (f(x_j) - f(x_i))) - g(f(x_i)) \\
&= g(f(x_j)) - g(f(x_i))
\end{aligned} \tag{7}$$

## M  VISUALIZATION FOR MISCLASSIFIED EXAMPLE FOR CIFAR-10

We provide the visualization for examples misclassified by our SMT + KNN model. In Figure 7, the figure is divided into two panels, with 25 images on the left and 24 images on the right. In the left panel, we show an image being classified on the top left corner and its 24 nearest neighbors. In Figure 7, the image being classified is an image of a "dog". Almost all of its 24 nearest neighbors are "horses." As a result, the "dog" image is misclassified as a "horse". The right panels show the patches with the highest cosine similarity to their corresponding patch in each of the 24 nearest neighbors. One insight we can learn from these visualizations is that these misclassifications are largely due to local-part matching. For example, a "deer" may look like a "horse" except for the "deer horn" parts. However, "deer horn" parts may only contribute a small fraction in the similarity measurement. As a result, the nearest neighbors of a "deer" may be almost all "horses". To solve this issue, again, we need to have a better understanding of hierarchical representation learning and high-level concept formation.

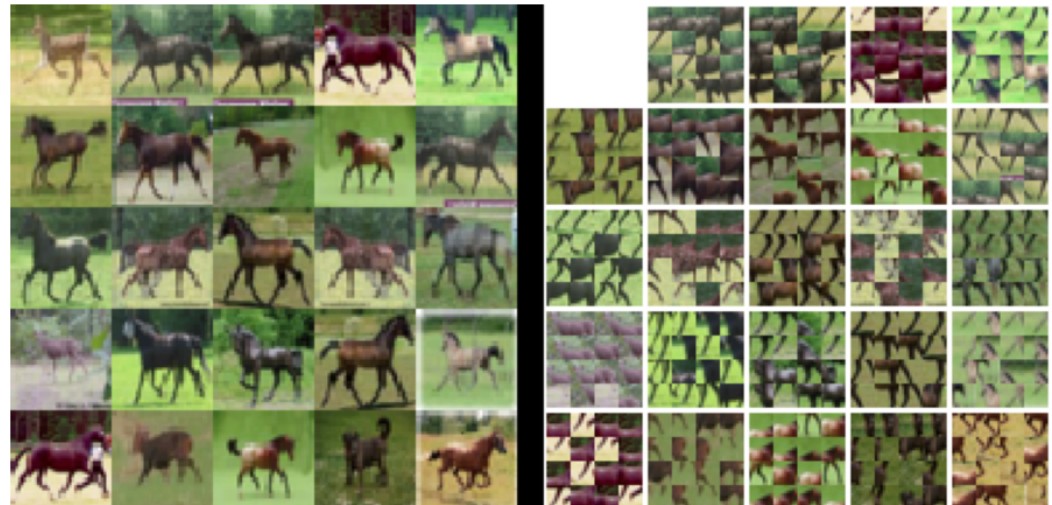

Figure 7: Misclassified example 1. A "deer" image is misclassified as horse.

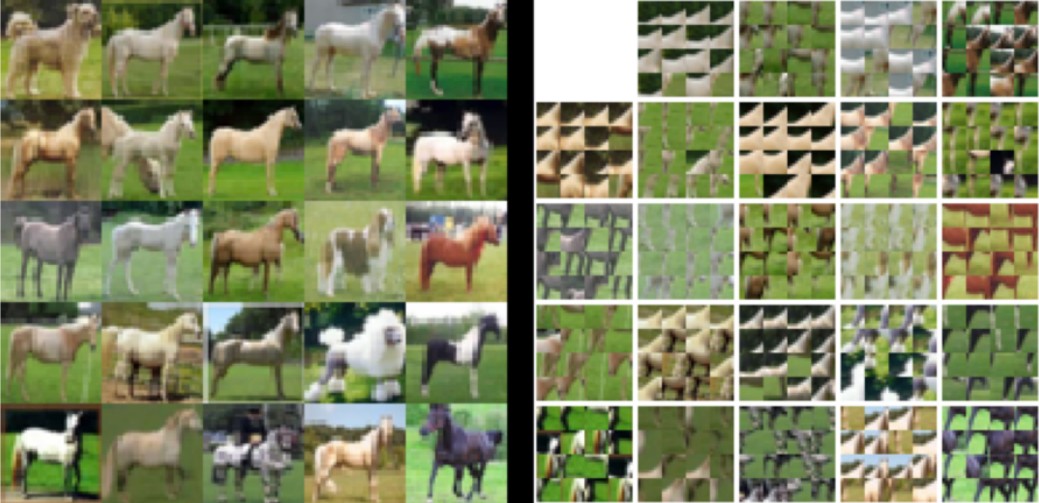

Figure 8: Misclassified example 2. A "dog" image is misclassified as horse.

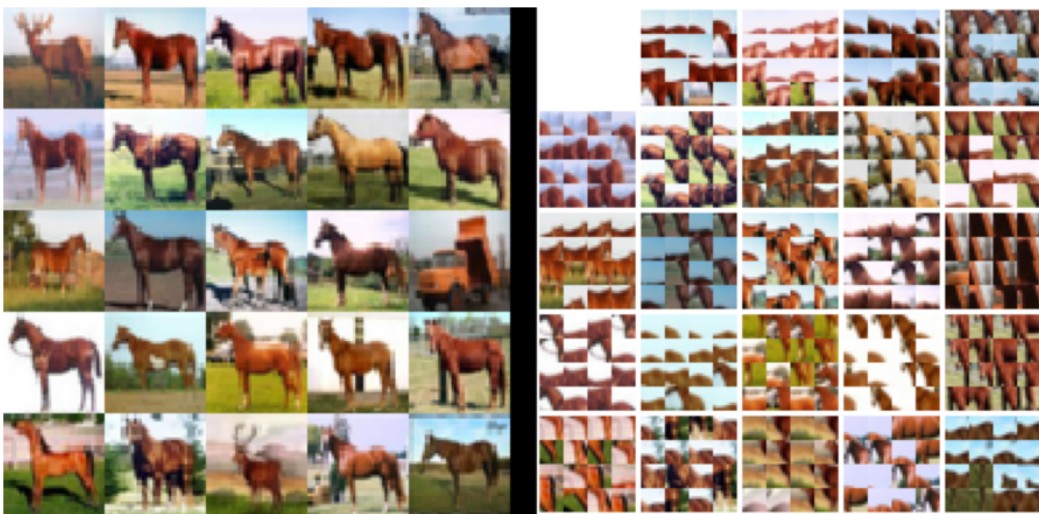

Figure 9: Misclassified example 3. A "deer" image is misclassified as horse.

