# OpenReview forum: "Minimalistic Unsupervised Representation Learning with the Sparse Manifold Transform"
_ICLR.cc/2023/Conference — ICLR 2023 notable top 25%_

### Official Review · Reviewer_hKPC · 2022-10-13

**Confidence:** 4
**Correctness:** 3
**Technical Novelty And Significance:** 3
**Empirical Novelty And Significance:** 4
**Recommendation:** 8

**Clarity, Quality, Novelty And Reproducibility:**

The paper is clear in terms of its claims and the experimental results back up the claims. I believe the experiments are reproducible. The main novelty of the paper is building a minimal model for unsupervised representation learning using SMT (which is an existing method). Compared to the SOTA SSL methods, they show that this method performs well. Furthermore, this minimal model is to be seen as a simplified version of VICReg, thus leading to better understanding of methods like VICReg which I believe is interesting and novel. However, for the last point, I believe the authors provide a more elaborate explanation. Overall, this is a good paper that I am inclined to accept.

**Details Of Ethics Concerns:**

No ethical concerns.

**Strength And Weaknesses:**

Strengths:

1. The paper studies an important problem which is to provide some understanding of SOTA SSL methods, and proposes to use SMT which is a simple model using a single-level dictionary learning (which is well understood) and a single linear spectral embedding (which is also well understood in manifold learning literature).

2. The paper is written well with ample explanations, connections with plenty of related work and illustration on a toy dataset.

3. Experiments on MNIST and CIFAR show results that to me are surprising. SMT performs comparably with SOTA SSL methods (when some data augmentations are turned off).

Weaknesses:

1. I don't quite understand why SMT cannot work with all data augmentations. Perhaps the authors can explain this point better.

2. Beyond empirical analysis, I am not seeing how this minimal setup is helping explain SOTA SSL methods. If I look at SimCLR, for example, the features are derived at the image level using augmentations as positives and other images as negatives. I don't see that happening with SMT. So, I would appreciate a clearer explanation as to how having this minimal SMT model helps understand SimCLR or VICReg.

3. I think related to (2), maybe a longer discussion on how to connect SMT and VICReg optimization and loss functions more concretely can help. I am not very familiar with VICReg, so I think this would definitely help me.

4. Besides SimCLR and VICReg, there are several other SSL models such as Masked Autoencoders, does SMT also help in understanding these models?

5. Also, are the authors saying that if there is a way to easily stack more layers of SMT in order to learn hierarchical representations, the gap between SMT and VICReg (for example) would not exist?

**Summary Of The Paper:**

The paper describes a barebones version of unsupervised representation learning framework based on the sparse manifold transform (SMT), which according to the authors, helps in understanding SOTA self-supervised representation learning (SSL) methods (such as VICReg). SMT has two steps: (1) sparse coding to map into a high-dimensional space and (2) manifold learning: a low-dimensional embedding that takes neighborhood/co-occurrence structure into account. Both sparsity (with respect to a dictionary) and local neighborhoods/co-occurrences serve as important features of natural signals like images, and even words (the sparsity is trivial).  While SMT is not a contribution of this paper, they show that SMT produces representations that capture the essence of unsupervised representation learning (SMT is a minimal model in this sense that it is a minimal model that takes both sparsity and neighborhood structure into account). The authors make connections with SOTA by pointing out that a deep neural encoder takes the place of sparse coding and the SMT optimization is very similar to that of VICReg. In principle, even SMT could be stacked in order to produce hierarchical representations that are necessary for larger natural images. Experiments on small-sized images in CIFAR illustrate that even SMT can produce representation that rival the SOTA.


**Summary Of The Review:**

I think the paper uses SMT which is a simple and easy-to-understand model in a novel setting in order to gain understanding of SSL methods, which is an interesting and important research direction. Experimental results on the small image datasets are convincing and do show that SMT is indeed a strong model in spite of being simple, which is great. However, I have noted some weaknesses that I hope the authors can respond to. Overall, this is a nice paper with interesting ideas and results and I am happy to increase my score based on the authors' response.

UPDATE AFTER AUTHOR RESPONSE:

I greatly appreciate the authors taking the time to significantly edit and improve the paper. Appendix K in particular helps me understand the connection between SMT and VICReg a lot better, and the loss functions indeed are similar. The minimal model does help illustrate that the ideas of SMT are useful in learning good SSL features. How these ideas fully translate to deep models trained with SGD with data augmentations is not clear, but this seems to be a good step in that direction. Perhaps if it is possible to train shallow (a few layers (?)) VICReg and showing the learned features are similar to SMT features in some sense, the connection would be even stronger. Overall, I am increasing my rating and recommending acceptance of the paper.

---

> ### Author Response · Authors · 2022-11-19
> **Reply to Reviewer hKPC (Part 1/2)**
>
> __Q1__: “ .. why SMT cannot work with all data augmentations ..”
>
> __A1__: The sparse feature function and the spectral embedding depend on the data. With a sparse feature function, the spectral embedding training takes only one epoch because it is deterministic rather than hundreds of epochs in the SOTA deep SSL methods. We didn’t find a recipe to efficiently apply augmentation like color jittering in SMT with our current sparse feature function. For example, a naive approach to do color augmentation would be: for each dictionary element, we add a color-jittered version of it back into the dictionary, and we repeat this multiple times. This will make our dictionary extremely large, which is inefficient, but we saw a slight performance gain. The deep SSL method compresses infinitely many color-jittered versions of an image into a neural network. We hope a competitive performance can be achieved without these various color augmentations. This is one of our future directions.
>
> ---
>
> __Q2__: “Beyond empirical analysis .. how this minimal setup is helping explain SOTA SSL methods .. SimCLR, for example, the features are derived at the image level using augmentations as positives and other images as negatives .. how having this minimal SMT model helps understand SimCLR or VICReg.”
>
> __A2__: This is a very important question. The SOTA joint-embedding has proved to be able to build an effective representation of images, which can be used for several downstream tasks. We want to answer why it works and how exactly such a representation is formed in an unsupervised fashion in high-dimensional space. On the one hand, we use reductionism to understand the theory of joint-embedding SSL. On the other, if we have a theory, we shall build minimalistic models based on the principles to achieve competitive performance. Two useful observations are:
> 1) The success of joint-embedding SSL is largely explained by a distributed representation of image patches. [1] One can use the SOTA joint-embedding methods to train on only fixed-scale small image patches to learn a representation of image patches. Aggregating the patch representation from all the patches from an image as the image representation achieves on-par performance as the baseline methods trained on multi-scale crops. This observation essentially shows that VICReg has the same loss function as the SMT objective in this work, as we mentioned at the end of Section 2 — Connection to the deep latent-embedding methods. We have also added Appendix-K to discuss this in more detail. In CIFAR, building a representation for 12x12 image patches is enough to achieve the baseline performance.
> 2) There exists a duality between the objective function from SimCLR and VICReg [2, 3], shown both algebraically and empirically. This observation shows that SimCLR and VICReg essentially have the same objective.
>
> Given these two observations, we can show that building a minimalistic model following the neural and statistical principles can achieve competitive performance compared to the SOTA joint-embedding SSL methods. Such a convergence of two seemingly distant approaches is surprising since we show that they follow the same optimization objective. This minimalistic model also provides a picture of how such a representation is formed in high-dimensional image-patch space. Please kindly refer to Figure 2 and Figure 4. The remaining gap between this work and the SOTA methods is that the SOTA method can handle higher-dimensional image patches, e.g., 12 x 12 patches in CIFAR10. Though we still do not understand why further closing this gap should lead to a better theory of joint-embedding SSL.
>
> [1] Chen et al. 2022. Intra-Instance VICReg: Bag of Self-Supervised Image Patch Embedding
>
> [2] Garrido et al. 2022. On the duality between contrastive and non-contrastive self-supervised learning
>
> [3] Balestriero and LeCun 2022. Contrastive and non-contrastive self-supervised learning recover global and local spectral embedding methods
>
> ---
>
> __Q3__: I think related to (2), maybe a longer discussion on how to connect SMT and VICReg optimization and loss functions more concretely can help. I am not very familiar with VICReg, so I think this would definitely help me.
>
> __A3__: Thanks for this suggestion! We agree that the connection between SMT and VICReg should be made more concrete. Thus, we added Appendix-K to explain the connection and make this paper more self-contained. To summarize, if both the sparse feature extraction and projection are parametrized as a neural network and learned through backpropagation. The SMT objective is the same as the VICReg objective. Please kindly refer to Appendix-K for a more detailed discussion.

---

> > ### Author Response · Authors · 2022-11-19
> > **Reply to Reviewer hKPC (Part 2/2)**
> >
> > __Q4__: Besides SimCLR and VICReg, there are several other SSL models such as Masked Autoencoders. Does SMT also help in understanding these models?
> >
> > __A4__: We do not really understand why masked autoencoders work or how the representation is formed in them. This is a big mystery for the whole SSL community. So far, we do not have a clear understanding of how the masked token prediction objective leads to a good latent representation. As a result, the latent representation in masked autoencoders is treated as an emerging phenomenon, which is yet universal in many different modalities. One way to draw a connection between masked autoencoders and joint-embedding SSL methods is to treat masking as a data-augmentation technique [4, 5]. In this case, masked augmentation used with Siamese network is strongly connected to joint-embedding SSL methods with typical data augmentation: insteading of trying to put the representation of two image patches close to each other, masked augmentation tries to put a set of unmasked image patches’ representation close to another set of unmasked image patches’ representation. SMT can also be formulated in this way. However, providing understanding for masked autoencoders is an important open problem.
> >
> > [4] Assran et al. 2022. Masked siamese networks for label-efficient learning
> >
> > [5] Li et al. 2022. Masked Siamese ConvNets
> >
> > ---
> >
> > __Q5__: Also, are the authors saying that if there is a way to easily stack more layers of SMT in order to learn hierarchical representations, the gap between SMT and VICReg (for example) would not exist?
> >
> > __A5__: There is some evidence that SMT layers can be stacked as shown in the original SMT paper [6]. However, we can not safely make this claim that simple stacking will close the gap. In fact, we suspect that simply stacking SMT will not fully close the gap, but it may help scale the current approach to high-dimensional datasets like ImageNet. Such scaling to ImageNet may lead to about 35%-40% KNN accuracy. As we mentioned in the discussion, we believe that the key to fully closing the gap between “white-box” models and the SOTA joint-embedding models is a better understanding of hierarchical representation and contextualized embedding, whereas stacking is only a coarse way of building hierarchical representation. Fully closing the gap and understanding hierarchical unsupervised representation learning points out an exciting opportunity for the theory of unsupervised representation learning.
> >
> > [6] Chen et al. 2018. The Sparse Manifold Transform.

---

### Official Review · Reviewer_77VK · 2022-10-21

**Confidence:** 3
**Correctness:** 4
**Technical Novelty And Significance:** 2
**Empirical Novelty And Significance:** 3
**Recommendation:** 8

**Clarity, Quality, Novelty And Reproducibility:**

The paper is clearly written and provide interesting explanations, examples and references from the literature. There are a few typos and misspellings in Part 3.

Up to my knowledge, this work presents new insights on the way SMT can be used for embedding.

As there is no source code, it would be nice to add some details on training procedures and hyper-parameters, especially for deep learning methods used as baselines. The paper may not be reproducible as is.

**Strength And Weaknesses:**

**Strengths**

The method proposed in the paper, which is an extension of SMT, allows to embed data from sparse coding and dictionary learning. The model is interesting because it is simple and seems to work well compared to SOTA Deep Learning methods on classical datasets. This might be of interest for people working on self-supervised learning and dictionary learning on various types of signals.

**Weaknesses**

- The contributions are not always clearly highlighted. I don’t understand whether Part 2 contains new ideas concerning SMT or not. In particular, the authors claim to “revisit the formulation of SMT” in the introduction. Could this be more specific ?

- The authors use 1-sparse sparse coding to learn the dictionary D and the representation codes in the experiments.  As described in the paper, this corresponds to performing a k-means algorithm. Figure 6 in appendix highlights that the number of atoms has to be extremely large (at least 8000 atoms of dimension 32) to achieve optimal performance, and I wonder if this is due to the 1-sparse constraint. Why focusing only on this case? This seems very restrictive, especially for complex data. It would be interesting to increase the number of non zero coefficients (even to consider relaxations of the $\ell_0$ constraint for efficiency purposes as it seems to be an issue for CIFAR datasets, see Mairal et al. 2009) to see whether this allows to reduce the dimension of the dictionary without impacting the performance in terms of knn accuracy and computation time (same kind of experiments as in Figure 6 in appendix). This might also help doing better than randomized dictionaries in CIFAR experiments. In my opinion, this study would improve the quality of the paper.

- There is no discussion about the impact of the number of samples on performance. Would it be possible (and interesting from the authors point of view) to integrate figures providing the knn accuracy with respect to the number of images to see whether SMT performs well with small datasets compared to baseline methods?

Minor remarks:
- In Table 2, the authors claim that “the MNIST patch space can be viewed as a 3-manifold”. I don’t understand how this can be inferred from the results shown in the table. Could the authors elaborate on that point ?
- There is not enough discussion about the impact of the context range, especially in the experiments on CIFAR, and it seems to be an important hyper-parameter of the model.

**Summary Of The Paper:**

The paper provides a method to learn an embedding from sparse coding and dictionary learning through the sparse manifold transform. The authors show that this “white-box model” performs well compared to Deep Learning methods on classical image datasets like MNIST and CIFAR.

**Summary Of The Review:**

Even though I think that the 1-sparse case is restrictive and that the paper lacks a bit of experimental grounding and details, the idea seems new and interesting in the context of SSL and dictionary learning, and is worth being highlighted.

------------------------------------

The authors added relevant details and experiments to the paper, and responded to my requests for clarification.

---

> ### Author Response · Authors · 2022-11-19
> **Reply to Reviewer 77VK (Part 1/2)**
>
> __Q1__: “I don’t understand whether Part 2 contains new ideas concerning SMT or not. In particular, the authors claim to “revisit the SMT” in the introduction.”
>
> __A1__: Thanks for the suggestion. We have revised wording accordingly. From “revisit the formulation of SMT” to “revisit the formulation of SMT with a more general perspective and discuss how it can be used to solve various unsupervised learning problems.”
>
> Section 2 is for several purposes: First, compared to the vast majority of the deep SSL methods, SMT is not as popular. So a concise description makes this paper self-contained. Second, we provide new explanations with concrete examples, a new co-occurrence point of view to complement the original manifold point-of-view, a more unified formulation to cover both similarity and linearity. Third, we also introduce how we can leverage SMT to build representation for benchmark purposes, which has been missing. In the following, we summarize the four contributions of this paper.
>
> 1) In SMT, the model's picture is dominated by a continuous manifold view. However, this work supplements this manifold point of view with a probabilistic co-occurrence point of view. The beauty is that we can capture both of these points of view with the same formulation.
> 2) SMT was introduced to combine the key ideas from three classical unsupervised learning methods: manifold learning, sparse coding, and slow feature analysis. How SMT was presented in the original paper may create a misleading image that time is necessary to establish this transformation. However, this is not true. In this paper, we provide a discussion that explains temporal locality is one type of locality out of the three classical localities: 1) Spatial locality, temporal locality, and neighborhoods in the raw signal space.
> 3) Though the SMT paper establishes the concept of the transform, the benchmark results were missing. In this work, we provide the benchmark results, which support the theoretical proposal in SMT.
> 4) This paper also provides a connection between SMT and VICReg (and other joint-embedding SSL methods as supported by [1, 2]). They share precisely the same optimization objective. On the one hand, SMT is constructed from neural and statistical principles. On the other hand, joint-embedding SSL methods are a major branch of SOTA methods. The convergence of these two seemingly distant approaches points towards a unified theory of unsupervised representation learning. And we share our opinion on how the remaining gap can be closed.
>
> [1] Balestriero et al. 2022. Contrastive and Non-Contrastive Self-Supervised Learning Recover Global and Local Spectral Embedding Methods
>
> [2] Garrido et al. 2022. On the duality between contrastive and non-contrastive self-supervised learning
>
> ---
>
> __Q2__: “The authors use 1-sparse coding to learn the dictionary D .. Figure 6 in appendix highlights that the number of atoms has to be extremely large (at least 8000 atoms of dimension 32) to achieve optimal performance, and I wonder if this is due to the 1-sparse constraint. Why focusing only on this case? ”
>
> __A2__: Though this is a slight misunderstanding, the reviewer has the right insight that 1-sparse feature function is less efficient in handling complex data space. As we have discussed in both Section 2 and Section 3, while 1-sparse feature $f_{vq}$ is sufficient for building SMT on MNIST patch space, it is less efficient in more complex patch space like CIFAR 10 space. So we have introduced a feedforward general sparse feature function $f_{gq}$. Empirically, this general sparse feature function performs close to inference-based sparse coding. But it has a benefit that $f_{gq}$ is much faster in benchmarking. Please kindly refer to Table 3 for a comparison between $f_{vq}$ and $f_{gq}$. $f_{vq}$ does not scale to colored CIFAR patches efficiently. In Appendix E and Figure 6, we mentioned that the sparse feature used was $f_{gq}$, which is not 1-sparse. But we agree that this should be made more clear. So we have revised accordingly with more clarification to avoid potential confusion. Even with general sparse features, we still find a high dimensional feature function is required to achieve the performance. As we mentioned, we have also tried inference-based sparse coding as used in [1], and the performance is comparable to that of $f_{gq}$.

---

> > ### Author Response · Authors · 2022-11-19
> > **Reply to Reviewer 77VK (Part 2/2)**
> >
> > __Q3__: “Would it be possible (and interesting from the authors point of view) to integrate figures providing the KNN accuracy with respect to the number of images to see whether SMT performs well with small datasets compared to baseline methods?”
> >
> > __A3__: Sure! We performed an ablation study on the number of training samples and added the result to Appendix H. We also provide the results in the table below. In this ablation, we compared the SMT model with 8192 sparse features $f_{gq}$ and the SimCLR model with no color augmentation. The result shows SimCLR experienced a much larger performance drop than SMT when training with limited data. We will provide more ablations, e.g., trying different color data augmentation, testing higher dimensional sparse features, and comparing with other baselines in the next revision.
> >
> > | Number of Training Examples | 1000 | 2000 | 4000 | 8000 | 16000 | 32000 | 50000 |
> > | :-----------:| :-----------: | :-----------: | :-----------: | :-----------:| :-----------: | :-----------: | :-----------: |
> > | SMT-GQ (8192, no color aug) |    58.7 %  |    63.7 %  |   67.2 %   |    70.7%  |   74.0  %   |  77.4  %  |  79.4%  |
> > | SimCLR (ResNet18, no color aug) | 28.9 % |  33.9 % |  37.8 % |  41.3  %   |  50.1 %    |   60.4  %  |   68.3%   |
> > ---
> >
> > __Q4__: Minor remark: “the MNIST patch space can be viewed as a 3-manifold [...] Could the authors elaborate on that point ?”
> >
> > __A4__: Great question! As we can see that in a 4-dimensional patch embedding space, the KNN evaluation accuracy can still achieve 98.8%. Then in a 3-dimensional patch embedding space, the accuracy quickly drops to 98.3%. Significant numerical drops like this are typically used in manifold learning to determine the essential degree of freedom of the data. So in this case, we hypothesize that this change may imply that the MNIST patch space is approximately a 3-manifold since the 4-dimensional patch embedding is L2 normalized and on a 3-sphere. This question helped us realize that the original statement might be too strong. So we decided to tune down the statement in the revision. Please see the revised Table 2 caption.
> >
> > ---
> >
> > __Q5__: Minor remark: “There is not enough discussion about the impact of the context range, especially in the experiments on CIFAR.”
> >
> > __A5__: As suggested by the reviewer, we provide an additional ablation study on the impact of the context range on CIFAR-10. Generally, a larger context range in CIFAR10 leads to slightly better performance, and we hypothesize that this is due to the co-occurrence point of view being more important on CIFAR 10 than MNIST patch space. As we discussed earlier, the MNIST patch space is closer to a low-dimensional manifold.
> >
> > The result is added to Appendix G and also shown below. In summary, as context length increases, the performance increases slightly for CIFAR-10. For CIFAR-10, the images have lots of high-level structures. For example, in an image of an animal, the animal’s “leg” and “head” might lie in two image patches far apart. But it is essential to capture that “leg” and “head” co-occur because this is a unique trait for animals, requiring the context range to be large. A more rigorous study of the topology of CIFAR image patches is an interesting future direction.
> >
> > | Context Range ($d$) |    4  |    6  |   12   |   24  |   32 |
> > | :-----------:| :-----------: | :-----------: | :-----------: | :-----------:| :-----------: |
> > |  KNN Accuracy | 78.0 % |  78.5 % |  78.5  % |  79.0  %   |  79.4 %    |
> >
> > ---
> >
> > __Q6__: “As there is no source code, it would be nice to add some details on training procedures and hyper-parameters, especially for deep learning methods used as baselines.”
> >
> > __A6__: We have added all the hyperparameters to train the deep SSL baseline models in Appendix F. These hyperparameters and settings follow the default setting from the solo-Learn [3] GitHub repo, except that we modify color augmentation for ablation study purposes. The empirical results section provided all the details for how to construct SMT representation for CIFAR and MNIST datasets. We will also release the source code for this paper for the camera-ready version to ensure the reproducibility of the key results.
> >
> > [3] https://github.com/vturrisi/solo-learn

---

> > > ### Comment · Reviewer_77VK · 2022-11-22
> > > **Answer to the authors**
> > >
> > > I thank the authors for the many details and new interesting experiments they added to the paper, and for the clarifications on novelty.
> > >
> > > Concerning sparse coding, there was indeed a misunderstanding as I thought the algorithm in CIFAR experiments was based on a 1-sparse feature function with a dictionary randomly sampled from data, where as it is based on cosine similarity between patches and a dictionary randomly sampled from data. To clarify my thoughts, I wanted to know whether learning the dictionary $D$ (with reduced size, for instance a few hundreds components) and sparse representation $\alpha$ (with more than 1 non zero element) with online or stochastic solvers (as proposed in Mairal et al. 2009) would perform better than randomly sampling the dictionary and be sufficiently time-efficient for the task.
> > >
> > > In any case, I updated my recommendation as the authors responded to my requests for clarification and added relevant experiments to the paper.

---

### Official Review · Reviewer_dX2g · 2022-10-24

**Confidence:** 4
**Correctness:** 3
**Technical Novelty And Significance:** 3
**Empirical Novelty And Significance:** 3
**Recommendation:** 6

**Clarity, Quality, Novelty And Reproducibility:**

*Clarity:* I feel that the paper uses quite a few buzzwords without defining their precise meaning (or at least to me these meanings were pretty unclear) in the paragraph “The sparse manifold transform”. For example on p3 “Similarity can be considered as a first-order derivative.”: A derivative of what w.r.t to what? Other examples: What is a “path in patch space”? What is a feature space “where locality and decomposition are respected”? In particular the I’m not sure if I understand what decomposition means in the context of image patches.

*Quality:* Please see strengths and weaknesses above.

*Novelty:* I could not find any references that apply the SMT to the image classification problems considered in this paper. However, the SMT is proposed in prior work, so the novelty is somewhat limited.

*Reproducibility:* Most of the details are explained well. However, I could not find details about the mentioned softmax-KNN classifier (e.g. how many neighbors are used?). Furthermore, as far as I can tell the fact that the EM algorithm is used to learn the 1-sparse feature representation is only stated in the appendix, and it would be useful to make this clear earlier. Finally, when applying GloVe to MNIST, how were the images tokenized?


*Typos:*
- p2: “representation” instead of “re-presentation”
- p4: “d-dimensional vector” instead of “d-dimensional vectors”
- p7: “colorjitter” instead of “colorgitter”


**Strength And Weaknesses:**

I think the overall direction of the paper is interesting and exploring ways to construct simple, interpretable models performing competitively with highly-overparameterized, very large models is a worthwhile endeavor. In more detail, I see the following strengths and weaknesses.

*Strengths:*
- The method is quite simple and produces surprisingly good results given its simplicity.
- The paper makes a good effort in describing the SMT and giving an intuitive understanding using the manifold disentanglement problem (Figure 2).
- The paper presents a number of interesting ablations.

*Weaknesses:*
- As acknowledged by the authors KNN, the reliance of their method on KNN limits the scalability of the method. This could be discussed in more detail: What does this mean in terms of compute and memory requirements? Also, have the authors considered training a small model (linear classifier, MLP) on top of the SMT representation? This could lead to more interesting compute/memory vs accuracy tradeoffs. Also what would be the effect of increasing resolution? The datasets considered by the authors have tiny images.
 -The paper could make a better case for interpretability. Instead of just looking at patches with similar embeddings, what are the nearest neighbors for misclassified examples? Which patches led to misclassification? Is there anecdotal evidence of the SMT allowing us to understand incorrect classifications?
- There are a number of prior works investigating engineered, principled transforms as a substitute for deep end-to-end learned black-box methods. These usually evaluate on MNIST and CIFAR-10/100, so comparing performance and methods would be possible. For example the scattering transform [a] and kernel methods [b].


[a] Oyallon, Edouard, and Stéphane Mallat. "Deep roto-translation scattering for object classification." Proceedings of the IEEE Conference on Computer Vision and Pattern Recognition. 2015.

[b] Shankar, Vaishaal, et al. "Neural kernels without tangents." International Conference on Machine Learning. PMLR, 2020.


**Summary Of The Paper:**

The paper studies the application of the Sparse Manifold Transform (SMT) to image classification problems. The SMT is a shallow transform that first sparsely represents the input data/image in a high-dimensional feature space (via a non-linear mapping) and then linearly embeds these sparse representations in a low-dimensional space. The authors work out the details of how to apply the SMT to image data and evaluate the method on MNIST and CIFAR-10/100, showing good results. In addition, some of the design choices are ablated.

**Summary Of The Review:**

The paper makes an interesting attempt at applying the SMT to image classification and shows quite strong results. The overall direction of the paper seems relevant and interesting to me.

The clarity of the paper could be improved, scaling aspects should be discussed in more detail, and the method should be compared to more related work.

---

> ### Author Response · Authors · 2022-11-19
> **Reply to Reviewer dX2g (Part 1/4)**
>
> __Q1__: “.. KNN limits the scalability of the method.” “.. have the authors considered training a small model (linear classifier, MLP) on top of the SMT representation?”
>
> __A1__: This is a good suggestion. KNN is not necessary. We use KNN to evaluate the representation because we believe a good representation should directly reflect “similarity” in the representation space without using labels to train an extra linear layer. KNN evaluation is typically a more challenging evaluation scheme than linear probing in self-supervised learning literature. Another reason we do not use linear probing is that even though linear probing is just one-layer training, we frequently find that suboptimal parameters can have a significant impact on evaluation performance, e.g., ~ 3%. Since this work is trying to avoid hyperparameter tuning as much as possible, we didn’t choose this evaluation.
>
> As suggested by the reviewer, we provide a comparison of the linear probing evaluation vs KNN evaluation for SMT in the following table. We use SMT-GQ with 8192 dictionary elements, i.e., the first layer has 8192 dimensions. No color augmentation is used in this ablation.
>
> | Embedding Dim | 50 | 110 | 210 | 350 | 500 | 800 | 1600 | 3200 | 6400 | 8192 |
> | :-----------:| :-----------: | :-----------: | :-----------: | :-----------:| :-----------: | :-----------: | :-----------: | :-----------: | :-----------: | :-----------: |
> | KNN Acc |    74.7 %  |    77.6 %  |   78.6 %   |   79.3 %  |   79.4 %   |   79.3 %  |    79.0 %  |    77.9 %  |      76.1%  |     75.2%  |
> |  Linear Probing | 69.0 % |  74.5 % |  77.0 % |  78.6 %   |  79.7 %    |   80.2 %  |   80.8 %   |   81.0 %   |      80.5 % |    80.4 %  |
>
> As we can see, when the patch embedding dimension is at 500, linear probing is slightly better than the KNN. With higher dimensional embedding, linear probing would perform better than KNN evaluation by up to 1.6 %. The linear probing result could be further improved with better hyperparameter tuning, which is beyond the scope of this work.
>
> ---
>
> __Q2__: What would be the effect of increasing resolution?
>
> __A2__: Though this is speculative, given our experience, we believe that this “white-box” model based on neural and statistical principles will probably achieve around 35-40% top1 KNN or linear-probing accuracy with ImageNet. To further close the gap, we suspect the key is a better understanding of hierarchical representation and contextualized embedding. A simple stacking of the representation may be enough to achieve 35-40%. But, to match the SOTA methods, a much clearer picture of hierarchical representation is needed. This is a fundamental problem that has been open for quite a long time. So we do not have a clear answer. This is an exciting direction for our future research, as discussed in the paper.
>
> ---
>
> __Q3__: “The paper could make a better case for interpretability .. what are the nearest neighbors for misclassified examples? Which patches led to misclassification? Is there anecdotal evidence of the SMT allowing us to understand incorrect classifications?”
>
> __A3__: Thanks for this suggestion! Per the request, we have included appendix-M, which is dedicated to answering these questions. Let’s take the examples misclassified as “horse.” One insight we can learn from these visualizations is that these misclassifications are largely due to local-part matching. E.g., a “deer” may look like a “horse” except for the “deer horn” parts. However,  “deer horn” parts may only contribute a small fraction in the similarity measurement. As a result, the nearest neighbors of a “deer” may be almost all “horses.” To solve this issue, again, we need to have a better understanding of hierarchical representation learning and high-level concept formation.

---

> > ### Author Response · Authors · 2022-11-19
> > **Reply to Reviewer dX2g (Part 2/4)**
> >
> > __Q4__: “There are a number of prior works investigating engineered, principled transforms as a substitute for deep end-to-end black-box methods .. comparing performance and methods would be possible.”
> >
> > __A4__: In this work, we directly compare with the SOTA SSL methods, which are more challenging in general. Please note that we have cited the literature on PCANet, ReduNet, scattering transform, and neural tangent kernel. As we mentioned at the beginning of this paper, the goal of the unsupervised representation in this work is that “similarity” shall be reflected in the representation space directly. KNN evaluation should directly lead to competitive performance. We are unaware of any other “white-box” model that can achieve a KNN performance comparable to this work.
> > Given the reviewer’s suggestion, we make a comparison with scatting transform [1] [2] as the following. Our reproduction of [1] is worse than that reported number in the original paper. So, we directly cite the highest CIFAR10 result in [1] as a comparison, denoted as RT-Scattering. In [1], many techniques are applied to improve the performance, such as rotation stable wavelet transform, averaging features from different orders of scattering transforms, and Orthogonal Least Square (OLS) feature reduction. Please also note that [1] uses supervised feature reduction and Gaussian SVM classifier, which is close to an optimized linear probing setting. In comparison, the linear projection layer in SMT is learned in a completely unsupervised fashion. However, SMT’s KNN result is already slightly better than the highest result in [1]. SMT’s linear probing shall achieve 1.5% - 2% additional top1 accuracy, as shown in the earlier linear probing ablation.
> > We also implemented a standard translation invariant scattering transform [2] using Kymatio, an off-the-shelf package [3]. We evaluated it with a soft KNN classifier, denoted at TI-Scattering. Further, we combine SMT with scattering transform by building a spectral embedding layer on top of the scattering features, denoted as TI-Scattering + SMT. As we can see, SMT can significantly improve the KNN accuracy of TI-Scattering by 18%.
> >
> > | RT-Scattering (LP) | SMT-GQ (KNN) | TI-Scattering (KNN) | TI-Scattering + SMT (KNN)|
> > | :-----------:| :-----------: | :-----------: | :-----------: |
> > | 82.3 % | 83.2% | 52.0 % | 70.1 %|
> >
> > There is a recent work on building joint-embedding self-supervised learning with neural tangent kernels [4]. SMT generally achieves significantly better results than [5] in terms of KNN evaluation accuracy. However, there is a deep connection between kernel methods and the sparse features. Let's take the heat kernel as an example. The kernel provides a way to respect the locality of the original signal space. Sparse features also provide a means to respect the locality. Building spectral embedding in either the kernel space or the sparse feature space will lead to a qualitatively similar behavior of shaping the distance.
> >
> > However, we would like to emphasize that we do not wish to claim superiority over any of these methods. They have different objectives. Potentially, they can be combined to build better representations. Together, they provide a more complete picture of the signal representation. And further works are needed to better understand the connections between them. All of the newly suggested references are cited in the revision.
> >
> > [1] Oyallon et al. 2015. Deep roto-translation scattering for object classification.
> >
> > [2] Mathieu et al. 2018. Kymatio: Scattering Transforms in Python.
> >
> > [3] Andén and Mallat 2013. Deep scattering spectrum.
> >
> > [4] Vaishaal Shankar et al. 2020. "Neural kernels without tangents."
> >
> > [5] Kiani et al. 2022. Joint Embedding Self-Supervised Learning in the Kernel Regime
> >
> > ---
> >
> > __Q5__:  What is a “path in the patch space”? This question refers to Figure 1(b).
> >
> > __A5__: Similarity can be considered as a first-order directional derivative of the representation transform w.r.t. the similarity neighborhoods. A representation transform $g$ maps a raw signal vector $\vec x$ into a representation embedding $\vec \beta$, i.e., $g: \mathbb{R}^m \to \mathbb{R}^d$. In SMT, $g = Pf(\cdot)$, where $f$ is a sparse feature function. Given a raw signal vector $\vec{x}_i$ and a raw signal vector $\vec{x}_j$ from $\vec{x}_i$’s similarity neighborhood $\\{ \vec{x}_j | j \in n(i) \\}$, the first-order directional derivative of $g$ is proportional to $g(\vec{x}_i) - g(\vec{x}_j)$, i.e., $P\vec{\alpha}_i - P\vec{\alpha}_j$. As we discussed at the beginning of the paper, “similarity” neighborhoods can be defined by 1) temporal co-occurrence, 2) spatial co-occurrence, 3) local metric neighborhoods in the raw signal space. For more detailed discussion, please kindly refer to Appendix L.

---

> > > ### Author Response · Authors · 2022-11-19
> > > **Reply to Reviewer dX2g (Part 3/4)**
> > >
> > > __Q6__:  What is a “path in the patch space”? This question refers to Figure 1(b).
> > >
> > > __A6__: Let’s take the manifold learning problem setting. The collection of all $m$-dimensional natural image patches occupies a nonlinear subset of $m$-dimensional Euclidean space. While different manifold learning methods use different techniques, they share the same basic idea: we can figure out the topology of the subset by leveraging local neighborhoods of each point in the subset. Such neighborhoods together define a neighborhood graph, which is made explicit in [6] and [7]. “A path in the patch space” means a path on this neighborhood graph. Though this continuous manifold picture is important, we believe that a co-occurrence point of view complements it significantly. So we provide the discussion in Figure 1(b) and the paragraph “Probabilistic point of view” in Section 2.
> > >
> > > [6] Tenenbaum et al. 2000. A Global Geometric Framework for Nonlinear Dimensionality Reduction
> > >
> > > [7] Saul and Roweis 2003. Think Globally, Fit Locally: Unsupervised Learning of Low Dimensional Manifolds
> > >
> > > ---
> > >
> > > __Q7__: What is a feature space “where locality and decomposition are respected”?
> > >
> > > __A7__: In manifold learning, the “locality is respected” means that Euclidean distance is respected locally. Figure 2 gives a concrete example. In Figure 2(b), given a set of dictionary elements (the set of dots), which tile the union of manifolds, a raw signal (red cross) is lifted by a 1-sparse feature function into a high dimensional space, shown as the red dot. This 1-sparse feature function faithfully respects the locality of the original raw signal space since the dictionary densely tiles the signal space.
> > >
> > > The 1-sparse feature function, denoted as $f_{vq}$ in the paper, is sufficient for simple signal space like the patches of MNIST, as we show with the MNIST benchmark. However, the 1-sparse feature function is harder to generalize to high-dimensional space, e.g., color-image space is much harder for the 1-sparse feature function than gray-scale image space. Further, for more complicated signal space, we need to consider decomposition. For example, even in a small CIFAR image patch, there can be several different “parts.”
> > >
> > > In this case, we need to go beyond the 1-sparse feature function and use the general sparse feature function, denoted as $f_{gq}$. A general sparse feature function shall be able to decompose a raw signal vector into a set of features. In sparse coding, this decomposition is solved by an inference, which is relatively time-consuming. An efficient implementation used in this paper is a feedforward general sparse feature function described in detail in Section 3. The general idea is that different sparse feature dimensions can identify different “parts” in a raw signal vector.
> > >
> > > __Q8__: “I could not find any references that apply the SMT to the image classification problems considered in this paper. However, the SMT was proposed in prior work, so the novelty is somewhat limited.”
> > >
> > > __A8__: Thanks for acknowledging that the application of SMT to these benchmarks is new. This is one of the four contributions of this work. In addition, there are three other contributions:
> > >
> > > 1) In SMT, the model's picture is dominated by a continuous manifold view. However, this work supplements this manifold point of view with a probabilistic co-occurrence point of view. The beauty is that we can capture both of these points of view with the same formulation.
> > > 2) SMT was introduced to combine the key ideas from three classical unsupervised learning methods: manifold learning, sparse coding, and slow feature analysis. How SMT was presented in the original paper may create a misleading image that time is necessary to establish this transformation. However, this is not true. In this paper, we provide a discussion that explains temporal locality is one type of locality out of the three classical localities: 1) Spatial locality, temporal locality, and neighborhoods in the raw signal space.
> > > 3) This paper also provides a connection between SMT and VICReg (and other joint-embedding SSL methods as supported by [1, 2]). In fact, they share precisely the same optimization objective. On the one hand, SMT is constructed from neural and statistical principles. On the other hand, joint-embedding SSL methods are a major branch of SOTA methods. The convergence of these two seemingly distant approaches points towards a unified theory of unsupervised representation learning. And we share our opinion on how the remaining gap can be closed.
> > >
> > > Compared to the vast majority of papers published in deep self-supervised learning, there is insufficient attention to this unique approach of building unsupervised representation based on neural and statistical principles. As we have discussed in the general reply, there is an exciting opportunity in this direction, and we hope to see many more papers focusing on this topic.

---

> > > > ### Author Response · Authors · 2022-11-19
> > > > **Reply to Reviewer dX2g (Part 4/4)**
> > > >
> > > > __Q9__: “.. I could not find details about the mentioned softmax-KNN classifier.”
> > > >
> > > > __A9__: Thanks for pointing this out! The soft-KNN classifier is frequently used in deep self-supervised learning literature without a detailed discussion. We apologize for omitting this. This KNN classifier is adopted from a soloLearn[8], which is a widely used SSL GitHub repo. Please refer to Appendix-J of the revision for a detailed description of the KNN classifier used in solo-learn.
> > > >
> > > > [8] https://github.com/vturrisi/solo-learn
> > > >
> > > > ---
> > > >
> > > > __Q10__:  “ .. EM algorithm is used to learn the 1-sparse feature representation is only stated in the appendix .. useful to make this clear earlier.”
> > > >
> > > > __A10__: As we mentioned in Section 3 (Empirical Results), 1-sparse feature learning is equivalent to clustering. At the same place, we also point the readers to the appendix for more details. It can be implemented with any clustering algorithm. The centroids of each cluster will serve as dictionary elements. We agree this can be made more clear. The paper has been revised accordingly. Please see the change in Section 3.
> > > >
> > > > ---
> > > >
> > > > __Q11__:   “.. applying GloVe to MNIST, how were the images tokenized?”
> > > >
> > > > __Q11__: Here, the tokenization is equivalent to 1-sparse feature, i.e., a 1-sparse feature can be converted to the corresponding index of non-zero entry of the 1-sparse feature (cluster centroid). In this case, the only difference between spectral embedding and GloVe embedding is how the embedding is calculated. In GloVe, the log co-occurrence matrix of dictionary elements is calculated for the whole dataset before we solve the matrix factorization problem.
> > > >
> > > > ---
> > > >
> > > > __Q12__: Typos
> > > >
> > > > __Q12__: Thanks! Fixed. “Re-presentation” was intentional to emphasize the transformation, but we agree with the reviewer that this may be confusing.

---

> > > > > ### Comment · Reviewer_dX2g · 2022-11-24
> > > > > **Response to author response**
> > > > >
> > > > > I thank the authors for their detailed response. I'm adapting my rating accordingly.

---

### Official Review · Reviewer_iJSh · 2022-10-24

**Confidence:** 3
**Correctness:** 1
**Technical Novelty And Significance:** 2
**Empirical Novelty And Significance:** 3
**Recommendation:** 6

**Clarity, Quality, Novelty And Reproducibility:**

The writing could be improved in several parts of the paper. For instance, the authors do not clearly state the contributions of the paper I introduction. Moreover, a large part of sections 1 and 2 focuses on the ideas of SMT which is method introduced in [20]. This makes sense to a certain extend for making the paper  self-contained. However, the authors should also make clear the key contributions of their work. Also, subsection ''SMT representation for MNIST and CIFAR" should be at the empirical results' section.

**Strength And Weaknesses:**

Sparse manifold transform is an approach proposed for signal representation in [20]. SMT is a white box method, meaning that it consists of interpretable steps i.e., sparse coding and low-dimensional spectral embedding. The authors build on SMT showing its efficiency in the manifold disentanglement problem. Moreover, they provide experimental results on MNIST, CIFAR10 and CIFAR100 showing promising performance of SMT as compared to SOTA SSL methods e.g. SimCLR and VICReg.

**Summary Of The Paper:**

The authors leverage the sparse manifold transform (SMT) and provide an insight into its merits on unsupervised learning. The authors show how SMT can be applied to the manifold disentanglement problem. As is pointed out, SMT as a white-box method can be an alternative to black box and self-supervised models building on the principles of parsimony namely sparsity and low-rank.

**Summary Of The Review:**

The authors leverage the sparse manifold transform showing its potential as a white-box unsupervised learning method. The main contribution of the paper the use of SMT in manifold disentanglement problem and its probabilistic point of view interpretation for co-occurence modeling. My main comments are the followings:

1) The overall presentation of ideas could be improved making clear the contributions of the current paper (see above).
2) The experimental results show promising performance of the methods on MNIST and CIFAR datasets. Do the authors believe that a similar performance could be obtained in more complicated and high-dimensional datasets such as IMAGEnet? Specifically, are there any ideas on how to efficiently learn the sparsifying transform in such cases? Also, how practical is the implementation of compositional models that the authors hint on the manuscript?
3) An  advantage of SMT as compared to alternative SSL methods is its white-box nature. What is the trade-off between the "simplicity" of the method and  modifications needed for making it efficient  in more complicated datasets such as IMAGEnet?


------------------------------
Post-rebuttal Update:
I want to thank the authors for their time and effort in responding to reviewers' comments and addressing their concerns. At this time, I will keep my scope the same.

---

> ### Author Response · Authors · 2022-11-19
> **Reply to Reviewer iJSh**
>
> __Q1__: “.. making clear the contributions of the current paper.”
>
> __A1__: This is a very important question. As we summarized in the general reply, there are four major contributions in this work:
> 1) In SMT, the model's picture is dominated by a continuous manifold view. However, this work supplements this manifold point of view with a probabilistic co-occurrence point of view. The beauty is that we can capture both of these points of view with the same formulation.
> 2) SMT was introduced to combine the key ideas from three classical unsupervised learning methods: manifold learning, sparse coding, and slow feature analysis. How SMT was presented in the original paper may create a misleading image that time is necessary to establish this transformation. However, this is not true. In this paper, we provide a discussion that explains temporal locality is one type of locality out of the three classical localities: 1) Spatial locality, temporal locality, and neighborhoods in the raw signal space.
> 3) Though the SMT paper establishes the concept of the transform, the benchmark results were missing. In this work, we provide the benchmark results, which support the theoretical proposal in SMT.
> 4) This paper also provides a connection between SMT and VICReg (and other joint-embedding SSL methods as supported by [1, 2]). They share precisely the same optimization objective. On the one hand, SMT is constructed from neural and statistical principles. On the other hand, joint-embedding SSL methods are a major branch of SOTA methods. The convergence of these two seemingly distant approaches points towards a unified theory of unsupervised representation learning. And we share our opinion on how the remaining gap can be closed.
>
> [1]Balestriero et al. 2022. Contrastive and Non-Contrastive Self-Supervised Learning Recover Global and Local Spectral Embedding Methods
>
> [2]Garrido et al. 2022. On the duality between contrastive and non-contrastive self-supervised learning
>
> ---
>
> __Q2__: subsection ''SMT representation for MNIST and CIFAR" should be at the empirical results' section
>
> __A2__: Representation construction with image patch representation is a part of the method. Inspired by this suggestion, we changed the name of the corresponding paragraph from “SMT representation for MNIST and CIFAR” to “SMT representation for natural images.”
>
> ---
>
> __Q3__: Do the authors believe that a similar performance could be obtained in more complicated and high-dimensional datasets like ImageNet? What is the trade-off between the “simplicity” of the method and modifications needed for making it efficient in more complicated datasets such as ImageNet?
>
> __A3__: Though this is speculative, given our experience, we believe that this “white-box” model based on neural and statistical principles will probably achieve around 35-40% top1 KNN or linear-probing accuracy with ImageNet. To further close the gap, we suspect the key is a better understanding of hierarchical representation and contextualized embedding. Simple stacking of the representation may achieve 35-40%. But, to match the SOTA methods, a much clearer picture of hierarchical representation is needed. This is a fundamental problem that has been open for quite a long time, and we do not have a clear answer. Given a better-unsupervised learning theory, we hope there will be no trade-off. Besides the learning accuracy can be improved, the more dramatic improvement shall come from the optimization. Since SMT only takes one training epoch to converge rather than hundreds of epochs used by the SOTA methods, we believe that we can leverage the understanding and improve the SOTA SSL training efficiency by orders of magnitude. This is an exciting direction for future research, as discussed in the paper.

---

> ### Author Response · Authors · 2022-12-06
> **On the Correctness**
>
> Thanks for the post-rebuttal reply! We just realized that the correctness score is 1. Are there additional issues we should address to alleviate the concerns on correctness? Thanks again!

---

### Author Response · Authors · 2022-11-19
**General Reply**

We thank our reviewers for their encouraging comments, helpful suggestions, and insightful questions. R1, R3, and R4 recommend accepting our paper based on white-box model construction, the new insights, and the surprising results. R2 also finds our results interesting and surprising. R1 iJSh says, “.. SMT as a white-box method can be an alternative to black box and self-supervised models building on the principles ..”; R2 dX2g says, “.. produces surprisingly good results given its simplicity .. giving an intuitive understanding ..”; R3 77VK says, “.. clearly written and provide interesting explanations, examples .. ” R4 hKPC says, “studies an important problem .. written well with ample explanations, connections with plenty of related work .. show results that to me are surprising.” The questions and suggestions help us revise the paper to deliver key messages better. Several reviewers suggest we highlight the contributions of this work, which we address first in this general reply. Then we also provide an overview of the revision alongside a revised paper. The added parts are marked as blue in the revision, whereas strike lines mark the removed parts.

This paper has four major contributions:
1) In SMT, the model's picture is dominated by a continuous manifold view. However, this work supplements this manifold point of view with a probabilistic co-occurrence point of view. The beauty is that we can capture both of these points of view with the same formulation.

2) SMT was introduced to combine the key ideas from three classical unsupervised learning methods: manifold learning, sparse coding, and slow feature analysis. How SMT was presented in the original paper may create a misleading image that time is necessary to establish this transformation. However, this is not true. In this paper, we provide a discussion that explains temporal locality is one type of locality out of the three classical localities: 1) Spatial locality, temporal locality, and neighborhoods in the raw signal space.

3) Though the SMT paper establishes the concept of the transform, the benchmark results were missing. In this work, we provide the benchmark results, which support the theoretical proposal in SMT.

4) This paper also provides a connection between SMT and VICReg (and other joint-embedding SSL methods as supported by Balestriero et al. 2022 and Garrido et al 2022). They share precisely the same optimization objective. On the one hand, SMT is constructed from neural and statistical principles. On the other hand, joint-embedding SSL methods are a major branch of SOTA methods. The convergence of these two seemingly distant approaches points towards a unified theory of unsupervised representation learning. And we share our opinion on how the remaining gap can be closed.

Here we provide an overview of the changes made in the revision:

1. A concise summary of the contributions is added to Section 1.
2. In Section 2: The sparse manifold transform, we clarified the expression for the differential operator $D$ for optimization (1).
3. In Section 2: SMT representation for MNIST and CIFAR, we changed the title from “SMT representation for MNIST and CIFAR” to “SMT representation for natural images.”
4. In Section 3, we clarified how 1-sparse sparse coding can be learned. We also clarified  how the evaluation of SMT on MNIST supports the hypothesis that MNIST patch space can be viewed as a 3-manifold.
5. Overall, we made some minor wording changes for clarification. We also fixed the typos suggested by reviewer dX2g.
6. In Appendix F, we provided the hyperparameter we used to train a deep SSL baseline for reproducibility purposes.
7. In Appendix G, we added an ablation study on the impact of context range for SMT, which provides insight into why both the manifold learning view and co-occurrence view are crucial.
8. In Appendix H, we added an ablation study on the impact of the number of training examples for SMT and compared it to a deep SSL baseline.
9. In Appendix I, we provided a comparison of the linear probing evaluation vs KNN evaluation for SMT.
10. In Appendix J, we provided details on the KNN algorithm we used for evaluation.
11. In Appendix K, we provided details on how the objective of VICReg can be explicitly formulated into SMT objectives.
12. In Appendix L, we clarified how to interpret similarity loss as a first-order directional derivative.
13. In Appendix M, we added visualizations for misclassified examples for CIFAR-10.
14. Nine additional references were added.

Next, we reply to each reviewer to address the specific questions.

---

### Decision · Program_Chairs · 2023-01-20

**Decision:**

Accept: notable-top-25%

**Justification For Why Not Higher Score:**

* Still some gap to be bridged with SOTA approaches

**Justification For Why Not Lower Score:**

* Unanimous support of the reviewers (and confidently so)
* Provide valuable insights to better understand SOTA SSL approaches
* Simple!
* Insightful discussions in the light of the related literature
* Likely to inform future research

**Metareview: Summary, Strengths And Weaknesses:**

The reviewers and meta reviewer all carefully checked and discussed the rebuttal. They thank the authors for their response and their efforts during the rebuttal phase. The response helped resolve many important concerns (e.g., clarification of the contributions, add study of linear probing, comparisons with manually-engineered transforms, evaluation in the low-sample regime).

The reviewers and meta reviewer all acknowledge that the submission proposes a simple (yet effective) and relevant approach to better understand fundamental components of SOTA SSL methods. The submission features as well a rich related work discussion and numerous insightful ablation studies (further improved post rebuttal).

All in all, the submission is recommended for acceptance.

**Note From Pc:**

if the above contains the word "oral" or "spotlight" please see: "oral" presentation means -> notable-top-5% and "spotlight" means -> notable-top-25%. As stated in our emails, we are disassociating presentation type from AC recommendations